# Learning with Exact Invariances in Polynomial Time

## Abstract

We study the statistical-computational trade-offs for learning with exact invariances (or symmetries) using kernel regression over manifold input spaces. Traditional methods, such as data augmentation, group averaging, canonicalization, and frame-averaging, either fail to provide a polynomial-time solution or are not applicable in the kernel setting. However, with oracle access to the geometric properties of the input space, we propose a polynomial-time algorithm that learns a classifier with *exact* invariances. Moreover, our approach achieves the same excess population risk (or generalization error) as the original kernel regression problem. To the best of our knowledge, this is the first polynomial-time algorithm to achieve exact (not approximate) invariances in this context. Our proof leverages tools from differential geometry, spectral theory, and optimization. A key result in our development is a new reformulation of the problem of learning under invariances, as optimizing an infinite number of linearly constrained convex quadratic programs, which may be of independent interest.

## 1 Introduction

While humans can readily observe symmetries or invariances in systems, it is generally challenging for machines to detect and exploit these properties from data. The objective of machine learning with invariances is to develop approaches that enable models to be trained and utilized under the symmetries inherent in the data. This framework is broadly applicable across various domains in the natural sciences and physics, including atomistic systems (Grisafi et al., 2018), molecular wavefunctions and electronic densities (Unke et al., 2021), interatomic potentials (Batzner et al., 2022), and beyond (Batzner et al., 2023). While many applications involve Euclidean symmetries (Smidt, 2021), the scope of such methods extends well beyond them to other geometries (Bronstein et al., 2017).

Learning with invariances has a longstanding history in machine learning (Hinton, 1987; Kondor, 2008). In recent years, there has been significant interest in the development and analysis of learning methods that account for various types of invariances. This surge in interest is strongly motivated by many models showing considerable success in practice. Empirical evidence suggests the existence of algorithms that can effectively learn under invariances while exhibiting strong generalization and computational efficiency. However, from a theoretical perspective, much of the focus has been on the expressive power of models, generalization bounds, and sample complexity. There remains a relative lack of understanding regarding the statistical-computational trade-offs in learning under invariances, even in foundational settings such as kernel regression.

Kernels, which have been among popular learning approaches, offer both statistical and computational efficiency (Scholkopf & Smola, 2018). Symmetries can be included in kernel learning in various ways. An immediate solution for learning with invariances seems to be *data augmentation* over the elements of the group. Moreover, most kernel-based approaches to learning with invariances rely on *group averaging*, a technique that involves summing over group elements. However, the typically large size of the group can make both of these approaches computationally prohibitive, even super-exponential in the dimension of input data. Alternative approaches, such as *canonicalization* and *frame averaging*, also suffer from issues like discontinuities and scalability challenges (Dym et al., 2024).

In light of these challenges, this paper seeks to address the following question:

> Is it possible to obtain an invariant estimator for the kernel regression problem that exhibits both strong generalization capabilities and computational efficiency?

The first contribution of this work is a detailed study of the problem of learning with invariances in the context of kernel methods. We argue that, while group averaging fails to produce exactly invariant estimators within a computationally efficient timeframe, alternative algorithms can generate invariant estimators for the kernel regression problem in time that is polylogarithmic in the size of the group. In other words, we demonstrate that it is possible to achieve an invariant estimator that is both computationally efficient and exhibits strong generalization. At first glance, this result may seem counterintuitive and even impossible, since it implies that enumerating all possible invariances is not required to design statistically efficient learning algorithms. This provides theoretical support for the empirical observation that computational efficiency and strong generalization are attainable in learning with invariances. To the best of our knowledge, this is the first algorithm that is both statistically and computationally efficient for learning with invariances in the kernel setting,

Learning with invariances can be formulated as a *nonconvex optimization* problem, which is not tractable directly. To design an efficient algorithm, we leverage the spectral theory of the Laplace-Beltrami operator on manifolds. Notably, since this operator commutes with all (isometric) group actions on the manifold, it is possible to find an orthonormal basis of Laplacian eigenfunctions such that each group action on the manifold acts on the eigenspaces of the Laplacian via orthogonal matrices. This theoretical framework allows us to reformulate the original problem of learning with invariances on manifolds as a *infinite collection of finite-dimensional convex quadratic programs*—one for each eigenspace—each *constrained by linear conditions*. By truncating the number of quadratic programs solved, we can efficiently approximate solutions to the primary nonconvex optimization problem, thereby approximating kernel solutions to the problem of learning with invariances. This reformulation not only enables us to derive a polynomial-time algorithm for kernel regression under invariances, but it may also have broader applications, the exploration of which we defer to future research.

Finally, we emphasize again that this work is centered on achieving *exact* invariance, as many applications—especially neural networks with strong empirical performance—are explicitly designed to incorporate exact invariances by construction. In summary, this paper makes the following contributions:

- We initiate the exploration of statistical-computational trade-offs in the context of learning with exact invariances, focusing specifically on kernel regression over manifold-structured input spaces.
- We reformulate the problem of learning under invariances in kernel methods, leveraging differential geometry and spectral theory, and cast it as infinitely many convex quadratic programs with linear constraints, for which we derive an efficient solution in terms of time complexity. We trade off computational and statistical complexity by controlling the number of convex quadratic programs solved to obtain the estimator.
- We introduce the first polynomial algorithm for learning with invariances in the general setting of kernel regression over manifolds.

## 2  RELATED WORK

Generalization bounds and sample complexity for learning with invariances have been extensively studied, particularly in the context of invariant kernels. Works such as Elesedy (2021), Bietti et al. (2021), Tahmasebi & Jegelka (2023), and Mei et al. (2021) provide insights into this area. Additionally, studies on equivariant kernels (Elesedy & Zaidi, 2021; Petrache & Trivedi, 2023) further our understanding of how equivariances affect learning. PAC-Bayesian methods have also been applied to derive generalization bounds under equivariances (Behboodi et al., 2022). More recently, Kiani et al. (2024) explored the complexity of learning under symmetry constraints for gradient-based algorithms. For studies on the optimization of kernels under invariances, see Teo et al. (2007).

A variety of methods have been proposed to enhance the performance of kernel-based learning models. One prominent approach is the use of random feature models (Rahimi & Recht, 2007), which approximate kernels using randomly selected features. Low-rank kernel approximation techniques,

such as the Nyström method (Williams & Seeger, 2000; Drineas et al., 2005), have also been proposed to reduce the computational complexity of kernel methods; see also Bach (2013); Cesa-Bianchi et al. (2015). Divide-and-conquer algorithms offer another pontential avenue for kernel approximation (Zhang et al., 2013). Additionally, the impact of kernel approximation on learning accuracy is well-documented in Cortes et al. (2010).

Our work focuses on learning with invariances, which differs significantly from the tasks of learning invariances or measuring them in neural networks. For example, Benton et al. (2020) address how neural networks can learn invariances, while Goodfellow et al. (2009) study methods to measure the degree of invariance in network architectures.

Invariance in kernel methods is not limited to group averaging. Other approaches such as frame averaging (Puny et al., 2022), canonicalization (Kaba et al., 2023; Ma et al., 2024), random projections (Dym & Gortler, 2024), and parameter sharing (Ravanbakhsh et al., 2017) have also been proposed to construct invariant function classes. However, canonicalization and frame averaging face challenges, particularly concerning continuity, which has been addressed in recent works like Dym et al. (2024).

In specialized tasks such as graphs, image, and pointcloud data, Graph Neural Networks (GNNs) (Scarselli et al., 2008; Xu et al., 2019), Convolutional Neural Networks (CNNs) (Krizhevsky et al., 2012; Li et al., 2021), and Pointnet (Qi et al., 2017a;b) have demonstrated the effectiveness of leveraging symmetries. Symmetries have also been successfully integrated into generative models (Biloš & Günnemann, 2021; Niu et al., 2020; Köhler et al., 2020). For a broader discussion on various types of invariances and their applications across machine learning tasks, see Bronstein et al. (2017).

## 3 BACKGROUND AND PROBLEM STATEMENT

**Notation.** We begin by establishing some frequently used notation. Let $\mathcal{M}$ be a smooth, compact, and boundaryless $d$-dimensional Riemannian manifold. The uniform distribution over the manifold is the normalized volume element corresponding to its metric. We denote the space of square-integrable functions over $\mathcal{M}$ by $L^2(\mathcal{M})$ and the space of continuous functions by $C(\mathcal{M})$. Furthermore, $H^s(\mathcal{M})$ represents the Sobolev space of functions on $\mathcal{M}$ with parameter $s$, defined as the set of functions with square-integrable derivatives up to order $s$. Larger values of $s$ correspond to greater smoothness, and it holds that $H^s(\mathcal{M}) \subseteq C(\mathcal{M})$ if and only if $s > d/2$, a condition we will assume throughout this paper. For each $n \in \mathbb{N}$, we define $[n] := \{1, 2, \ldots, n\}$. We use $\log$ to denote the logarithm with base 2. We refer to Appendices A.1 and A.2 for a quick review of Riemannian manifolds.

**Problem statement.** We consider a general learning setup on a smooth, compact, and boundaryless Riemannian manifold $\mathcal{M}$ of dimension $d$. Our objective is to identify an estimator $\widehat{f} \in \mathcal{F}$ from a feasible space of estimators $\mathcal{F} \subseteq L^2(\mathcal{M})$, based on $n$ independent and uniformly distributed labeled samples $\mathcal{S} = \{(x_i, y_i) : i \in [n]\} \subseteq (\mathcal{M} \times \mathbb{R})^n$ drawn from the manifold. Here, the labels $y_i$ for $i \in [n]$ are produced based on the (unknown) ground truth regression function $f^\star \in C(\mathcal{M})$, meaning that $y_i = f^\star(x_i) + \epsilon_i$, for each $i \in [n]$, where $\epsilon_i, i \in [n]$, is a sequence of independent zero-mean random variables with variance bounded by $\sigma^2$. The population risk (or generalization error) of an estimator $\widehat{f} \in L^2(\mathcal{M})$, which quantifies the quality of the estimation, is defined as:

$$\mathcal{R}(\widehat{f}) := \mathbb{E}\left[\|\widehat{f} - f^\star\|^2_{L^2(\mathcal{M})}\right],$$

where the expectation is taken over the randomness of the data and labels.

Given a dataset of size $n$, finding estimators with minimal population risk can be quite complex, often requiring the resolution of non-convex optimization objectives. However, in scenarios where $f^\star \in \mathcal{H}$, with $\mathcal{H} \subseteq L^2(\mathcal{M})$ being a Reproducing Kernel Hilbert Space (RKHS), it is feasible to compute kernel-based estimators with low risk efficiently. Specifically, the Kernel Ridge Regression (KRR) estimator for the RKHS $\mathcal{H} = H^s(\mathcal{M})$, denoted as $\widehat{f}_{\mathrm{KRR}}$, achieves a population risk of $\mathcal{R}(\widehat{f}_{\mathrm{KRR}}) = \mathcal{O}\left(n^{-s/(s+d/2)}\right)$ while being computable in time $\mathcal{O}(n^3)$, assuming access to an oracle that computes the kernel associated with the space. Note that Sobolev spaces $H^s(\mathcal{M})$ with $s > d/2$ are RKHS. We refer the reader to Appendices A.8 and A.9 for a detailed review of the KRR estimator and related topics on Sobolev spaces.

**Learning with invariances.** We assume that a finite group $G$ acts smoothly and isometrically[1] on the manifold $\mathcal{M}$, represented by a smooth function $\theta : G \times \mathcal{M} \to \mathcal{M}$ mapping the product manifold $G \times \mathcal{M}$ to $\mathcal{M}$. We employ the notation $\theta(g, x)$ as $gx$ for any $g \in G$ and $x \in \mathcal{M}$. In a scenario of learning under invariances, the regression function $f^\star$ is invariant under the action of the group $G$, satisfying $f^\star(gx) = f^\star(x)$ for each $g \in G$ and $x \in \mathcal{M}$. Thus, learning under invariances introduces an additional requirement: not only must we compute an estimator with minimal population risk efficiently, but $\widehat{f}$ must also be invariant with respect to $G$. This additional condition is often satisfied in neural network applications by constructing networks that are invariant *by design*, such as graph neural networks.

In the context of learning with Sobolev kernels, the KRR estimator $\widehat{f}_{\mathrm{KRR}}$ is *not* $G$-invariant (see Appendix A for more details). Consequently, the KRR estimator cannot provide a solution for learning under invariances. However, with a shift-invariant Positive Definite Symmetric (PDS) kernel[2] $K : \mathcal{M} \times \mathcal{M} \to \mathbb{R}$, one can utilize *group averaging* to derive a new kernel and a new RKHS holding only $G$-invariant functions:

$$K_{\mathrm{inv}}(x_1, x_2) \coloneqq \frac{1}{|G|} \sum_{g \in G} K(gx_1, x_2).$$

Given that the Sobolev space $H^s(\mathcal{M})$ adopts a shift-invariant PDS kernel (Appendix A.8), one can apply the above method to construct and compute a $G$-invariant kernel (assuming access to evaluating its original kernel). This indicates that the KRR estimator on $K_{\mathrm{inv}}$ yields an invariant estimator for $f^\star$ with a desirable population risk (see Tahmasebi & Jegelka (2023) for a comprehensive study).

However, in terms of computational complexity, this method requires $\Omega(n^2 |G|)$ time to compute the new kernel between pairs of input data. In many practical scenarios, $|G|$ can be intolerably large. For instance, for the permutation group $P_d$, we have $|G| = d! \sim \sqrt{d}(\frac{d}{e})^d$ which is super-exponential in $d$. Consequently, the group averaging method cannot provide an efficient algorithm for learning with exact invariances. We emphasize "exact invariance" here, as the sum involved in $K_{\mathrm{inv}}$ can be approximated by summing over a number of random group transformations. However, this does not guarantee exact invariance, which is the primary goal of this paper.

Other traditional approaches to achieving learning under invariances include data augmentation, canonicalization, and frame averaging. For data augmentation, we need to increase our dataset size by a multiplicative factor of $|G|$, which is often impractical within efficient time constraints. This is because for any datapoint $x_i \in \mathcal{S}$, new datapoint $gx_i$ for any group element $g \in G$ should be added to dataset to ensure invariance of the underlying learning procedure in a blackbox way, leading to $\Omega(n|G|)$ complexity. Canonicalization involves mapping data onto the quotient space of the group action and subsequently finding an estimator (e.g., a KRR estimator) on the reduced input space. However, this method is also infeasible for kernels due to the unavoidable discontinuities and non-smoothness of the canonicalization maps, which violate RKHS requirements (Dym et al., 2024). Finally, frame averaging is analogous to canonicalization, but it remains unclear how to address continuity issues for efficient frame sizes. Moreover, it requires careful design of frames tailored to the specific problem at hand, making it unsuitable for a general-purpose algorithm. Thus, motivated by these observations, we pose the following question:

> Is it possible to obtain a $G$-invariant estimator for $f^\star \in H^s(\mathcal{M})$ with a desirable population risk (similar to the case without invariances) in $\mathrm{poly}(n, d, \log(|G|))$ time?

We aim to answer this question affirmatively in the following section. This is surprising, as it suggests that even enumerating the set $G$ is not required to find statistically efficient $G$-invariant estimators.

---

[1]The assumption of isometric action is made for simplicity; the proof can be extended to non-isometric actions using standard techniques in the literature, as discussed in Tahmasebi & Jegelka (2023).

[2]A kernel $K : \mathcal{M} \times \mathcal{M} \to \mathbb{R}$ is termed shift-invariant with respect to group $G$ if and only if $K(gx_1, gx_2) = K(x_1, x_2)$ for each $g \in G$ and $x_1, x_2 \in \mathcal{M}$. Shift-invariant kernels are *not* necessarily $G$-invariant. For example, one can show that $H^s(\mathcal{M})$ adopts a shift-invariant kernel while still producing non-invariant functions in its RKHS. We cover the details in Appendix A.9.

**Oracles.** To characterize computational complexity, first we need to specify the type of oracle access provided for the estimation. Before doing so, we briefly review the spectral theory of the Laplace-Beltrami operator on manifolds. For further details, we refer the reader to Appendix A.

The Laplace-Beltrami operator generalizes the Laplacian operator to Riemannian manifolds. It has a basis of smooth eigenfunctions $\phi_{\lambda,\ell} \in L^2(\mathcal{M})$, which serve as an orthonormal basis for $L^2(\mathcal{M})$. The index $\lambda$ represents the eigenvalue corresponding to the eigenfunction $\phi_{\lambda,\ell}$, and $\ell \in [m_\lambda]$ runs over the multiplicity of $\lambda$, denoted by $m_\lambda$. The eigenvalues can be ordered such that $0 = \lambda_0 < \lambda_1 \leq \lambda_2 \leq \cdots \to \infty$. For example, in the case of the sphere $S^{d-1}$, the spherical harmonics, which are homogeneous harmonic polynomials, are a natural choice of eigenfunctions.

The sequence of eigenfunctions and their corresponding eigenvalues provide critical information about the geometry of the manifold. In this work, we make use of the following two types of oracles:

- The ability to evaluate any eigenfunction $\phi_{\lambda,\ell}(x)$ at a given point $x \in \mathcal{M}$.

- The ability to compute the $L^2(\mathcal{M})$ inner product between a shifted eigenfunction $\phi_{\lambda,\ell}(gx)$ and another eigenfunction $\phi_{\lambda,\ell'}(x)$ for any group element $g \in G$.

For both oracles, we assume free access as long as $D_\lambda := \sum_{\lambda' \leq \lambda} m_{\lambda'} = \mathrm{poly}(n, d)$, where $D_\lambda$ denotes the number of eigenfunctions with eigenvalues less than or equal to $\lambda$. This assumption is motivated by the case of $S^{d-1}$, where spherical harmonics can be efficiently evaluated or multiplied in low dimensions (in such cases, only a few monomials need to be processed, making the task simple[3]). The first oracle handles the geometric structure of the manifold, while the second oracle captures the relationship between the group action and the manifold's spectrum. Both are crucial for obtaining our results.

## 4 MAIN RESULT

In this section, we address the question raised in the previous section by presenting the primary result of the paper, which is encapsulated in the following theorem.

**Theorem 1** (Learning with exact invariances in polynomial time). *Consider the problem of learning with invariances with respect to a finite group $G$ using a labeled dataset of size $n$ sampled from a manifold of dimension $d$. Assume that the optimal regression function belongs to the Sobolev space of functions of order $s$, i.e., $f^\star \in H^s(\mathcal{M})$ for some $s > d/2$ and let $\alpha := 2s/d$. Then, there exists an algorithm that, given the data, produces an exactly invariant estimator $\widehat{f}$ such that:*

- *It operates in time $\mathcal{O}\big(\log^3(|G|)n^{3/(1+\alpha)} + n^{(2+\alpha)/(1+\alpha)}\big)$;*

- *It achieves an excess population risk (or generalization error) of $\mathcal{R}(\widehat{f}) = \mathcal{O}\big(n^{-s/(s+d/2)}\big)$;*

- *It requires $\mathcal{O}\big(\log(|G|)n^{2/(1+\alpha)} + n^{(2+\alpha)/(1+\alpha)}\big)$ oracle calls to construct the estimator;*

- *For any $x \in \mathcal{M}$, the estimated label $\widehat{f}(x)$ can be computed in time $\mathcal{O}\big(n^{1/(1+\alpha)}\big)$ using $\mathcal{O}\big(n^{1/(1+\alpha)}\big)$ oracle calls.*

The full proof of Theorem 1 is presented in Appendix B.3, while a detailed proof sketch is provided in Section 5, and the algorithm is outlined in Algorithm 1.

Let us interpret the above theorem. Note that without any invariances, the Kernel Ridge Regression (KRR) estimator (details are given in Appendix A.9) provides an estimator $\widehat{f}_{\mathrm{KRR}}$ for the Sobolev space $H^s(\mathcal{M})$ that is computed in time $\mathcal{O}(n^3)$ and achieves the risk $\mathcal{R}(\widehat{f}_{\mathrm{KRR}}) = \mathcal{O}\big(n^{-s/(s+d/2)}\big)$. Here, while KRR cannot guarantee an exactly invariant estimator, we propose another estimator which is both exactly invariant and also converges with the same rate $\mathcal{O}\big(n^{-s/(s+d/2)}\big)$. As a result, we achieve exact invariances with statistically desirable risk (or sample complexity). In other words, the population risk is the same as the optimal case without invariances, which shows that the algorithm introduces no loss in statistical performance while enforcing group invariances.

---

[3]This also extends to other settings, such as the Stiefel manifold or tori.

We thus came to the following conclusion:

> The problem of learning with exact invariances can be efficiently solved in time $\text{poly}(n, d, \log(|G|))$ and with excess population risk (or generalization error) $\mathcal{O}\big(n^{-s/(s+d/2)}\big)$ which is the same statistical performance as for learning without invariances.

It is worth mentioning that, according to the theorem, the proposed estimator $\widehat{f}$ is not only efficiently achievable but also efficiently computes new predictions on unlabeled data.

*Remark* 1. We notice that in the proof of Theorem 1, the actual time and sample complexity depends only on the size of minimum generating set of the group $G$, denoted by $\rho(G)$, instead of $\log(|G|)$. We use logarithm in the theorem just to give better insights to the reader about the improvement from the naive approach. Thus, the actual proof allows to even achieve a tighter result, since $\rho(G) \leq \log(|G|)$ for any finite group (see Proposition 5 in Appendix B.1). Note that for some cases (such as cyclic groups) $\rho(G) \ll \log(|G|)$.

## 5 ALGORITHM AND PROOF SKETCH

In this section, we provide a proof sketch for Theorem 1, introducing several new notations and concepts necessary for achieving the reduction in time complexity.

We begin with the most natural optimization program for obtaining an estimator: the Empirical Risk Minimization (ERM), which proposes the following estimator:

$$\widehat{f}_{\text{ERM}} := \arg\min_{f \in H^s(\mathcal{M})} \left\{ \frac{1}{n} \sum_{i=1}^{n} (f(x_i) - y_i)^2 \right\},$$

where $\mathcal{S} = \{(x_i, y_i) : i \in [n]\} \subseteq (\mathcal{M} \times \mathbb{R})^n$ denotes the sampled (labeled) dataset.

However, as discussed, this method does not necessarily produce an estimator that is exactly invariant. A natural idea is to introduce group invariances as constraints into the above optimization, leading to the following constrained ERM solution:

$$\widehat{f}_{\text{ERM-C}} := \arg\min_{f \in H^s(\mathcal{M})} \left\{ \frac{1}{n} \sum_{i=1}^{n} (f(x_i) - y_i)^2 \right\}$$
$$\text{s.t.} \quad \forall (g, x) \in G \times \mathcal{M} : f(gx) = f(x).$$

While this formulation ensures exact invariance, it introduces $|G|$ functional equations. This is problematic for two reasons: first, $|G|$ constraints are prohibitively many, and second, these constraints require solving functional equalities, which are not easily achievable. Moreover, the functional equations involve non-linear (pointwise) constraints on the estimator function, which at first glance appear intractable due to nonconvexity of the contraints $f(gx) = f(x)$ for general choice of $g$.

Therefore, it is necessary to reformulate the above optimization program. The goals of the reformulation are to reduce the number of constraints and encode the functional equations into more tractable constraints, ideally linear ones.

**Reducing the number of constraints.** We begin by using the following basic property (based on the group law):

$$\Big( \forall g \in \{g_1, g_2\}, \forall x \in \mathcal{M} : f(gx) = f(x) \Big) \implies \Big( \forall x \in \mathcal{M} : f(g_1 g_2 x) = f(x) \Big).$$

This observation allows us to eliminate many unnecessary constraints. Specifically, we only need constraints over a small subset of $G$ if this subset can generate any group element through arbitrary group multiplications. To formalize this, we introduce the following definition:

**Definition 1.** A finite group $G$ is said to be generated by a subset $S \subseteq G$ if for every $g \in G$, there exists a sequence of elements $s_1, s_2, \ldots, s_k$ such that for each $i \in [k]$, either $s_i \in S$ or $s_i^{-1} \in S$ and $g = s_1 s_2 \cdots s_k$. The minimum size of such a subset $S$ is denoted by $\rho(G)$.

Clearly, $\rho(G) \leq |G|$. However, it can be shown (see Appendix B.1) that $\rho(G) \leq \log(|G|)$, which represents an exponential improvement over the trivial upper bound.

Thus, we can reformulate the constrained ERM optimization as:

$$\widehat{f}_{\text{ERM-C}} := \underset{f \in H^s(\mathcal{M})}{\arg\min} \left\{ \frac{1}{n} \sum_{i=1}^{n} (f(x_i) - y_i)^2 \right\}$$
$$\text{s.t.} \quad \forall (g, x) \in S \times \mathcal{M} : f(gx) = f(x),$$

where $|S| \leq \log(|G|)$. In this way, we reduce the number of constraints from $|G|$ to $\log(|G|)$ by leveraging the concept of minimal group generators. Note that this fact cannot be directly used in data augmentation, group averaging, or canonicalization techniques.

**Optimization in the spectral domain.** The constrained ERM formulation presented above, while advantageous in terms of reducing the number of constraints, involves optimizing over the infinite-dimensional space $H^s(\mathcal{M})$, which is computationally intractable. One way to make this problem tractable is to parametrize the estimator and search for the optimal parameters. To achieve this, we utilize the spectral theory of the Laplace-Beltrami operator over manifolds. While a detailed discussion of spectral theory is provided in Appendix A, we summarize the relevant concepts here.

As mentioned earlier, the Laplace-Beltrami operator yields a sequence of orthonormal eigenfunctions $\phi_{\lambda,\ell} \in L^2(\mathcal{M})$, where $\lambda \in \{\lambda_0, \lambda_1, \ldots\} \subseteq [0, \infty)$ represents the eigenvalue corresponding to the eigenfunction $\phi_{\lambda,\ell}$, and $\ell \in [m_\lambda]$ indexes the multiplicity of $\lambda$, denoted by $m_\lambda$. Therefore, any estimator $f \in L^2(\mathcal{M})$ can be expressed as:

$$f(x) = \sum_{\lambda} \sum_{\ell=1}^{m_\lambda} f_{\lambda,\ell} \phi_{\lambda,\ell}(x), \quad f_{\lambda,\ell} := \langle f, \phi_{\lambda,\ell} \rangle_{L^2(\mathcal{M})}.$$

The idea is to parametrize the problem by finding the best coefficients $f_{\lambda,\ell}$. However, since there are infinitely many eigenvalues, there are infinitely many parameters to estimate, which is not feasible in finite time. Fortunately, we know that $f^\star \in H^s(\mathcal{M})$. From the definition of Sobolev spaces (see Appendix A.8), we have:

$$\|f^\star\|_{H^s(\mathcal{M})}^2 := \sum_{\lambda} \sum_{\ell=1}^{m_\lambda} (f_{\lambda,\ell}^\star)^2 D_\lambda^\alpha,$$

where $D_\lambda = \sum_{\lambda' \leq \lambda} m_{\lambda'}$, and $\alpha := \frac{2s}{d} > 1$.

Thus, we conclude that:

$$\sum_{\lambda : D_\lambda > D} \sum_{\ell=1}^{m_\lambda} (f_{\lambda,\ell}^\star)^2 \leq D^{-\alpha} \|f^\star\|_{H^s(\mathcal{M})}^2 = \mathcal{O}(D^{-\alpha}),$$

for any $D > 0$. This shows that for Sobolev regression functions $f^\star \in H^s(\mathcal{M})$, we can truncate the estimation of coefficients at a certain cutoff frequency $\lambda$, which allows the problem to be parametrized with finitely many parameters. Although this introduces bias into the estimation (since higher-frequency eigenfunctions will not be captured), the bias is bounded by the above inequality for Sobolev spaces.

Interestingly, this spectral approach yields a more meaningful optimization problem when considering the population risk function rather than ERM. The population risk, which is the primary objective in regression, is given by:

$$\mathcal{R}(f) = \mathbb{E}_{\mathcal{S}} \left[ \|f - f^\star\|_{L^2(\mathcal{M})}^2 \right] = \sum_{\lambda} \sum_{\ell=1}^{m_\lambda} \mathbb{E}[(f_{\lambda,\ell} - f_{\lambda,\ell}^\star)^2].$$

**Constrained spectral method.** To review, we introduced an efficient way to impose the constraints related to group invariances in the ERM objective and later presented spectral methods for obtaining estimators. The last step here is to combine these to achieve exact invariances via a constrained spectral method. We use an important property of the Laplace-Beltrami operator to introduce the algorithm.

Let $\Delta_{\mathcal{M}}$ denote the Laplace-Beltrami operator on the manifold $\mathcal{M}$, and let $G$ be a group acting isometrically on $\mathcal{M}$. Define the linear operator $T_g : f(x) \mapsto f(gx)$ for each group element $g \in G$ and any smooth function $f$ on the manifold. Then, we have

$$\Delta_{\mathcal{M}}(T_g \phi) = T_g(\Delta_{\mathcal{M}}(\phi)),$$

for any smooth function $\phi$ on the manifold (for a formal proof, please refer to Appendix A.5).

This identity tells us that the Laplace-Beltrami operator $\Delta_{\mathcal{M}}$ commutes with the operator $T_g$ for each $g$. Since both operators are linear, spectral theory implies that the commutativity shows the eigenspaces of $\Delta_{\mathcal{M}}$ are *preserved* under the action of the group $G$, meaning the operators can be simultaneously diagonalized. Specifically, for any $\lambda, \ell$, and any $g \in G$, the function $\phi_{\lambda,\ell}(gx)$ is a linear combination of eigenfunctions $\phi_{\lambda,\ell'}, \ell' \in [m_\lambda]$. In particular, the group $G$ acts via orthogonal matrices on the eigenspace $V_\lambda := \text{span}(\phi_{\lambda,\ell} : \ell \in [m_\lambda])$ for each $\lambda$.

Let $D^\lambda(g)$ denote the $m_\lambda \times m_\lambda$ orthogonal matrix corresponding to the action of an element $g \in G$ on $V_\lambda$ for each $\lambda$. Then, a function

$$f(x) = \sum_{\lambda} \sum_{\ell=1}^{m_\lambda} f_{\lambda,\ell} \phi_{\lambda,\ell}(x)$$

is $G$-invariant if and only if

$$D^\lambda(g) f_\lambda = f_\lambda, \quad \forall g \in G \, \forall \lambda \in \{\lambda_0, \lambda_1, \ldots\},$$

where $f_\lambda := (f_{\lambda,\ell})_{\ell \in [m_\lambda]} \in \mathbb{R}^{m_\lambda}$ for each $\lambda$. We can further reduce the number of conditions by passing $G$ to a generator set, which gives only $\log(|G|)$ conditions.

Thus, the commutativity of the Laplace-Beltrami operator and any isometric group action allows us to introduce only linear constraints on the spectral method to achieve exact invariances. This leads to the following optimization program:

$$\min_{f_{\lambda,\ell}} \sum_{\lambda} \sum_{\ell=1}^{m_\lambda} \mathbb{E}[(f_{\lambda,\ell} - f_{\lambda,\ell}^\star)^2],$$

$$\text{s.t.} \quad \forall g \in S \, \forall \lambda \in \{\lambda_0, \lambda_1, \ldots\} : D^\lambda(g) f_\lambda = f_\lambda.$$

Here, $f_{\lambda,\ell}^\star = \mathbb{E}_x[f^\star(x)\phi_{\lambda,\ell}(x)] = \mathbb{E}_{x,y}[y\phi_{\lambda,\ell}(x)]$ is not known a priori; only $n$ samples $(x_i, y_i) \in \mathcal{M} \times \mathbb{R}, i \in [n]$, are given. Furthermore, the constraints are independent for different eigenspaces (i.e., different $\lambda$), and the objective is a sum over eigenspaces. This means we can decompose the problem into a set of linearly constrained optimization programs, one for each eigenspace $V_\lambda$:

$$\min_{f_{\lambda,\ell}} \sum_{\ell=1}^{m_\lambda} \mathbb{E}[(f_{\lambda,\ell} - f_{\lambda,\ell}^\star)^2],$$

$$\text{s.t.} \quad \forall g \in S : D^\lambda(g) f_\lambda = f_\lambda.$$

This reformulation allows us to propose efficient estimators for the problem.

**Empirical estimator.** In this paper, we suggest the following auxiliary empirical mean estimator from the data for the above optimization program on $V_\lambda$:

$$\widetilde{f}_{\lambda,\ell} = \frac{1}{n} \sum_{i=1}^{n} y_i \phi_{\lambda,\ell}(x_i), \quad \forall \ell \in [m_\lambda]. \tag{1}$$

Moreover, we stop estimation and set $\widetilde{f}_{\lambda,\ell} = 0$ when $D_\lambda > D$, where $D$ is a hyperparameter. To find a $G$-invariant using our primary estimator, we solve the following quadratic program to find a solution satisfying the constraints for each $V_\lambda$ with $D_\lambda \leq D$:

$$\widehat{f}_{\lambda,\ell} := \arg\min_{f_{\lambda,\ell}} \sum_{\ell=1}^{m_\lambda} (f_{\lambda,\ell} - \widetilde{f}_{\lambda,\ell})^2,$$

$$\text{s.t.} \quad \forall g \in S : D^\lambda(g) f_\lambda = f_\lambda.$$

This optimization problem is a convex quadratic program with linear constraints that can be solved iteratively using the rich convex optimization machinery. Additionally, it has a closed-form solution as noted in Proposition 6 in Appendix B.2. Let $B^\lambda \in \mathbb{R}^{|S|m_\lambda \times m_\lambda}$ defined as the augmented matrix resulted by concatenating $D^\lambda(g) - I$ for all $g \in S$ on top of each other, i.e., $B^\lambda = [(D^\lambda(g_1) - I)^\top, (D^\lambda(g_2) - I)^\top, \dots, (D^\lambda(g_{|S|}) - I)^\top]^\top$. Then,

$$\widehat{f}_{\lambda,\ell} = \widetilde{f}_{\lambda,\ell} - {B^\lambda}^\top (B^\lambda {B^\lambda}^\top)^\dagger (B^\lambda \widetilde{f}_\lambda)[\ell],$$

where $\dagger$ denotes Moore–Penrose inverse.

The final estimator of the algorithm is given by

$$\widehat{f}(x) = \sum_{\lambda : D_\lambda \leq D} \sum_{\ell=1}^{m_\lambda} \widehat{f}_{\lambda,\ell} \phi_{\lambda,\ell}(x).$$

This meta approach to design $G$-invariant estimator $\widehat{f}$ from any primary estimtor $\widetilde{f}$ is novel and can be of independent interest. A pseudocode for this method is presented in Algorithm 1. Since the invariance is imposed in the spectral representation, we coin our proposed algorithm Spectral Averaging (`Spec-Avg`).

---

**Algorithm 1** Learning with Exact Invariances by Spectral Averaging (`Spec-Avg`)

**Input:** Input $\mathcal{S} = \{(x_i, y_i) : i \in [n]\}$ and $\alpha = 2s/d \in (1, \infty)$.

**Output:** Output $\widehat{f}(x)$.

1: Initialize $D \leftarrow n^{1/(1+\alpha)}$.
2: **for** each $\lambda$ such that $D_\lambda \leq D$ **do**
3:     **for** each $\ell \in [m_\lambda]$ **do**
4:         $\widetilde{f}_{\lambda,\ell} \leftarrow \frac{1}{n} \sum_{i=1}^n y_i \phi_{\lambda,\ell}(x_i)$.
5:     **end for**
6: **end for**
7: **for** each $\lambda$ such that $D_\lambda \leq D$ **do**
8:     Solve the following linearly constrained quadratic program over $m_\lambda$ variables:

$$\widehat{f}_{\lambda,\ell} \leftarrow \arg\min_{f_{\lambda,\ell}} \sum_{\ell=1}^{m_\lambda} (f_{\lambda,\ell} - \widetilde{f}_{\lambda,\ell})^2,$$

$$\text{s.t.} \quad \forall g \in S : D^\lambda(g) f_\lambda = f_\lambda.$$

9: **end for**
10: **return** $\widehat{f}(x) = \sum_{\lambda : D_\lambda \leq D} \sum_{\ell=1}^{m_\lambda} \widehat{f}_{\lambda,\ell} \phi_{\lambda,\ell}(x)$.

---

We conclude this section by reviewing how we apply the results from Algorithm 1 to the two following important examples.

**Example 1.** Consider the problem of learning under invariances over the unit sphere $S^{d-1} := \{x \in \mathbb{R}^d : \|x\|_2 = 1\}$, where the group $G$ is the group of all permutations of coordinates. Note that $|G| = d!$, which is prohibitively large for data augmentation or group averaging. However, this group is generated by only two elements: $\sigma_1 = (1\,2)$ and $\sigma_2 = (1\,2\,\dots\,d)$. Here, $\sigma_1$ swaps the first and second coordinates, while $\sigma_2$ is a cycle that maps $1 \to 2, 2 \to 3$, and so on, cyling with $d \to 1$.

The eigenspaces $V_\lambda$ for the sphere are precisely the sets of homogeneous harmonic polynomials of degree $k$, where $\lambda = k(k + d - 2)$. The permutation group acts on $V_\lambda$ by permuting the variables of the polynomials. This action is clearly linear, and the matrices $D^\lambda(g)$ can be efficiently computed (using tensor products) as long as $k$ is small. Moreover, homogeneous polynomials of degree $k$ can also be computed efficiently for small $k$. This shows that the oracles considered in this paper align perfectly with what we observe in the important case of spheres and polynomial regression. In Algorithm 1, we first compute the coefficients of each polynomial for degree $k$, up to a small $k$, and then solve a quadratic program with only two linear constraints to obtain an exactly invariant polynomial solution.

**Example 2.** Consider the same setup as the previous example but assume $d = 2$, i.e., the manifold is the unit circle. In this case, each eigenspace $V_\lambda$ is spanned by $\sin(k\theta)$ and $\cos(k\theta)$, where $\lambda = k^2$.

Let us assume our task is to find an estimator invariant with respect to rotations by integer multiples of $\frac{2\pi}{|G|}$. This group is cyclic and is generated by only one element $g_0 = \frac{2\pi}{|G|}$. Thus, we have only one constraint for each eigenspace. Indeed, one can observe that $D^\lambda(g_0) = R(k\frac{2\pi}{|G|})$, where $R(.) \in \mathbb{R}^{2\times 2}$ is the two-dimensional rotation matrix. Thus, this example further illustrates how our oracles are defined to solve the problem.

## 6 Discussion and Future Directions

We initiated the study on computational-statistical trade-offs in learning with exact invariances. We designed an algorithm that shows achieving the desirable population risk (the same as kernel regression without invariances) in $\text{poly}(n, d, \log(|G|))$ time for the task of kernel regression with invariances on general manifolds. We note that, for simplicity, we have focused on boundaryless manifolds and isometric group actions. However, using standard techniques, the theory can be extended to more general cases as well[4]. While the proposed spectral algorithm is computationally efficient, it does not offer any improvement in sample complexity over the baseline $\mathcal{R}(\widehat{f}) = \mathcal{O}\big(n^{-s/(s+d/2)}\big)$. It has been observed that without computational constraints, better convergence rates are possible for learning with invariances (Tahmasebi & Jegelka, 2023), which are minimax optimal. Thus, it remains open whether those improved rates are achievable in $\text{poly}(n, d, \log(|G|))$ time.

We note that the oracle access we assumed is primarily motivated by the case of the sphere, where polynomials can be evaluated, multiplied, composed by group elements, and integrated efficiently when they are of relatively low degree. We believe this is the most natural oracle access for this problem, as it aligns well with applications involving polynomials. An interesting future work could be to investigate the statistical-computational trade-offs using alternative oracles, e.g., similar to the kernel trick, how to design computationally efficient algorithms that have only access to the inner product of the RKHS. Another interesting future direction is to find whether random feature models as approximations for kernels can significantly improve the statistical-computational trade-off of learning with invariances. At present, our theory does not apply to random feature models.

We also observe that the spectral algorithm used in this paper does not employ the kernel trick, as it requires access to the entire set of features, rather than just their inner products. An interesting question is whether it is possible to utilize kernel tricks and find an alternative (polynomial-time) algorithm for learning under invariances. This approach could potentially improve the statistical efficiency of the spectral algorithm. In the end, we would like to note that capturing computational-statistical trade-offs in other estimation problems with invariances such as density estimation (Chen et al., 2023; Tahmasebi & Jegelka, 2024) could serve as a compelling avenue for future research.

## 7 Conclusion

In this paper, we explore the statistical-computational trade-offs in learning with invariances, focusing specifically on kernel regression. We observe that while the Kernel Ridge Regression (KRR) estimator can address this problem, it is not necessarily invariant without group averaging. Furthermore, since performing group averaging can be costly for large groups, we ask whether it is possible to develop statistically sound estimators with efficient time complexity. Our findings show that by reformulating the problem and reducing the number of constraints using group laws, we can express it as solving an infinite series of quadratic optimization programs under linear constraints. We conclude with an algorithm that achieves an exactly invariant estimator with polynomial time complexity and highlight several additional questions for future research.

## 8 Reproducibility and Ethics Statement

The main focus of this work is the theoretical understanding of computational-statistical trade-offs in learning with invariances; therefore, there are no significant ethical concerns. To ensure reproducibility, we have provided a detailed proof sketch of the algorithm in the main text. Additionally, all proofs and relevant background information are discussed in detail in the appendix.

---

[4]See e.g., Tahmasebi & Jegelka (2023).

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

# A  BACKGROUND

## A.1  RIEMANNIAN MANIFOLDS

In this section, we review some fundamental definitions from differential geometry and refer the reader to Lee (2006); Petersen (2006); Lee (2012) for further details.

**Definition 2** (Manifold). A topological *manifold* $\mathcal{M}$ of dimension $\dim(\mathcal{M})$ is a completely separable Hausdorff space that is locally homeomorphic to an open subset of Euclidean space of the same dimension, specifically $\mathbb{R}^{\dim(\mathcal{M})}$. More formally, for each point $x \in \mathcal{M}$, there exists an open neighborhood $U \subseteq \mathcal{M}$ and a homeomorphism $\phi : U \to \widehat{U}$, where $\widehat{U} \subseteq \mathbb{R}^{\dim(\mathcal{M})}$.

The value $\dim(\mathcal{M})$ is referred to as the *dimension* of the manifold. Examples of manifolds include tori, spheres, $\mathbb{R}^d$, and graphs of continuous functions. *Manifolds with boundaries* differ from boundaryless manifolds in that they may have neighborhoods that locally resemble open subsets of closed $\dim(\mathcal{M})$-*dimensional upper half-spaces*, denoted as $\mathbb{H}^{\dim(\mathcal{M})} \subseteq \mathbb{R}^{\dim(\mathcal{M})}$, defined as follows:

$$\mathbb{H}^d = \{(x_1, x_2, \ldots, x_d) \in \mathbb{R}^d \mid x_d \geq 0\}.$$

**Definition 3** (Local Coordinates). Given a *chart* $(U, \phi)$—a pair consisting of a local neighborhood $U$ and the corresponding homeomorphism $\phi : U \to \widehat{U}$—on a manifold $\mathcal{M}$ with dimension $d$, we define *local coordinates* $(x^1, x^2, \ldots, x^d)$ such that

$$\phi(p) = (x^1(p), x^2(p), \ldots, x^d(p)),$$

for each point $p \in U$.

**Definition 4** (Tangent Space). At each point $x \in \mathcal{M}$, the *tangent space* $T_x\mathcal{M}$ is defined as the vector space formed by the tangent vectors to the manifold $\mathcal{M}$ at $x$. A tangent vector $v \in T_x\mathcal{M}$ can be represented as the derivative of a smooth curve $\gamma(t) : (-\epsilon, \epsilon) \to \mathcal{M}$ defined on the manifold with the property that $\gamma(0) = x$. It is expressed as

$$\nu = \frac{d}{dt}\gamma(t)\Big|_{t=0}.$$

The tangent space $T_x\mathcal{M}$ is a real vector space with dimension $\dim(\mathcal{M})$.

**Definition 5** (Riemannian Metric Tensor). A *Riemannian metric tensor* $g$[5] on a manifold $\mathcal{M}$ is a smooth inner product defined on the tangent space $T_x\mathcal{M}$ at each point $x \in \mathcal{M}$. For any two tangent vectors $u, v \in T_x\mathcal{M}$, the metric assigns a real number $g_x(u, v) \in \mathbb{R}$.

**Definition 6** (Riemannian Manifold). A *Riemannian manifold* is defined as a pair $(\mathcal{M}, g)$, where $\mathcal{M}$ is a manifold and $g$ is a *Riemannian metric tensor* defined on the tangent space $T_x\mathcal{M}$ at each point $x \in \mathcal{M}$.

A Riemannian metric tensor provides essential tools for the study of manifolds, which we formalize below. It enables the following:

- the definition of the geodesic distance $d(x, y)$ between any two points $x, y \in \mathcal{M}$ on the manifold,
- a volume element $d\,\mathrm{vol}_g(x)$ over the manifold, serving as the measure for the Borel sigma-algebra over open subsets of the manifold $\mathcal{M}$, and
- the measurement of the angle between any two tangent vectors $u, v \in T_x\mathcal{M}$, which in turn provides the size of tangent vectors.

**Definition 7** (Geodesic Distance). The *geodesic distance* $d_\mathcal{M}(x, y)$ between any two points $x, y \in \mathcal{M}$ on the manifold is defined as the infimum length among all smooth curves $\gamma : [0, 1] \to \mathcal{M}$ connecting $x$ to $y$ ($\gamma(0) = x, \gamma(1) = y$). The length of a curve $\gamma$ is defined as

$$L(\gamma) = \int_0^1 \sqrt{g_{\gamma(t)}\left(\dot{\gamma}, \dot{\gamma}\right)}\, dt,$$

where $\dot{\gamma}$ denotes the derivative $\frac{d\gamma}{dt}$.

---

[5]This notation differs from $g$, which denotes group elements.

**Definition 8** (Volume Element). The *volume element* $d\mathrm{vol}_g(x)$ on a Riemannian manifold $(\mathcal{M}, g)$ is defined as

$$d\mathrm{vol}_g = \sqrt{\det(g_{ij})}\, dx^1 \wedge dx^2 \wedge \cdots \wedge dx^n,$$

where $g_{ij}$ are the components of the Riemannian metric tensor, $(x^1, x^2, \ldots, x^n)$ are the local coordinates, and $\wedge$ denotes the exterior product.

The volume element provides a way to compute the volume of subsets of $\mathcal{M}$ by integrating functions over $\mathcal{M}$. Moreover, a Borel measure $\mu$ on open subsets of $\mathcal{M}$ can be derived form the volume element to form probability measure space, e.g., uniformly over the manifold.

**Definition 9** (Smooth Map). A maping $f : \mathcal{M} \to \mathcal{N}$ is a smooth map if for any charts $(U, \phi)$ on $\mathcal{M}$, and $(V, \psi)$ on $\mathcal{N}$, the composition function $\psi \circ f \circ \phi^{-1} : \mathbb{R}^{\dim(\mathcal{M})} \to \mathbb{R}^{\dim(\mathcal{N})}$ is infinitely differentiable.

**Definition 10** (Pullback of the metric tensor). Given Riemannian manifolds $\mathcal{M}$, $(\mathcal{N}, g)$ and $\varphi : \mathcal{M} \to \mathcal{N}$ a smooth map between them. The *pullback of the metric tensor* $g$ by $\phi$, denoted by $\varphi^* g$ is the Riemannian metric tensor on manifold $\mathcal{M}$ defined by,

$$(\varphi^* g)_x(u, v) = g_{\varphi(x)}(d\varphi_x(u), d\varphi_x(v)), \text{ for all points } x \in \mathcal{M} \text{ and all } u, v \in T_x\mathcal{M},$$

where $d\varphi_x : T_x\mathcal{M} \to T_{\varphi(x)}\mathcal{N}$ is the differential of the map $\varphi$ at point $x$.

Thus, the pullback metric $\varphi^* g$ on $\mathcal{M}$ captures the relation between tangent vectors of $\mathcal{M}$ in terms of how they are mapped to the manifold $\mathcal{N}$ via $\varphi$.

**Definition 11** (Connected Manifold). A manifold $\mathcal{M}$ is *connected* if for any two points $x, x' \in \mathcal{M}$, there is a smooth curve $\gamma : [0, 1] \to \mathcal{M}$ such that $\gamma(0) = x$ and $\gamma(1) = x'$.

Throughout this paper, we focus on smooth, connected, compact and boundaryless Riemannian manifolds $(M, g)$ unless stated otherwise. For a Riemannian manifolds $(M, g)$, we denoted the dot product induced by the metric tensor $g$ as $\langle u, v \rangle_{g_x} = g_x(u, v)$ for all $u, v \in T_x$. We drop the subscript $x$ whenever it is clear from the context.

## A.2 Functional Spaces over Manifolds

Now equipped with probability measures on manifold discussed in Appendix A.1, we are ready to define functional spaces $L^p(\mathcal{M})$ and Sobolev spaces $\mathcal{H}^s(\mathcal{M})$ on manifold $\mathcal{M}$ analogously to their Euclidean counterparts in the following,

**Definition 12** (Functional Spaces on Manifolds). The Lebesgue functional spaces $L^p(\mathcal{M})$ for $p \in [1, \infty]$, and the Sobolev spaces $H^s(\mathcal{M})$ for $s \geq 0$ on a smooth Manifold $\mathcal{M}$, are defined as follows:

- The *Lebesgue space* $L^p(\mathcal{M})$ consists of measurable functions $f : \mathcal{M} \to \mathbb{R}$ such that $\|f\|_{L^p(\mathcal{M})} < \infty$ where,

$$\|f\|_{L^p(\mathcal{M})} = \begin{cases} \left( \int_{\mathcal{M}} |f(x)|^p \, d\mu(x) \right)^{1/p} & \text{if } p \in [1, \infty) \\ \mathrm{ess\,sup}_{x \in \mathcal{M}} |f(x)| < \infty. & \text{if } p = \infty \end{cases},$$

  where $\mu$ is the uniform measure over the manifold $\mathcal{M}$.

- The *Sobolev space* $H^s(\mathcal{M})$ consists of measurable functions whose derivatives up to order $s$ are in $L^2(\mathcal{M})$, i.e.,

$$H^s(\mathcal{M}) = \left\{ f \in L^2(\mathcal{M}) \mid D^\alpha f \in L^2(\mathcal{M}) \text{ for all multi-indices } \alpha \text{ with } |\alpha| \leq s \right\}.$$

## A.3 Lie Group of Isometry Maps

In this section, first we state basic definition of isometric mappings over manifolds and then wrap up by characterizing the isometry group over the manifold.

**Definition 13** (Isometry Map). A bijective mapping $\tau : \mathcal{M} \to \mathcal{M}$ is an *isometry* on the manifold $(\mathcal{G}, g)$ if $d(\tau(x), \tau(x')) = d(x, x')$.

We also state a brief definition of Lie groups for completeness.

**Definition 14** (Lie group)**.** A group $G$ is a *Lie group* with smooth group operations (multiplication and inversion) if it is additionally a smooth manifold.

The space of bijective Riemannian isometries defined on the manifold $(\mathcal{M}, g)$, denoted by $\mathrm{ISO}(\mathcal{M}, g)$ constitutes a group with composition operation. The celebrated Myers–Steenrod theorem states that any isometry map $\tau \in \mathrm{ISO}(\mathcal{M}, g)$ between connected manifolds is an isometry (Myers & Steenrod, 1939; Palais, 1957). Myers & Steenrod (1939) took it a step further and proved that isometry group of a Riemannian manifold $(\mathcal{M}, g)$ is a Lie group.

Alternatively, $\mathrm{ISO}(\mathcal{M}, g)$ can be charecterized by the pullback of the metric tensor. In terms, $\tau \in \mathrm{ISO}(\mathcal{M}, g)$ if and only if $g = \tau^* g$ (Petersen, 2006).

### A.4 Laplacian on Manifolds

In this section, we reiterate over definition of Laplace-Beltrami operator on manifolds (which is the generalization of the Laplacian operator $\Delta = \partial_1^2 + \partial_2^2 + \cdots + \partial_d^2$ defined on the Euclidean space $\mathbb{R}^d$) and state a several interesting properties that will utilize later. We refer to Chavel (1984) for additional details.

**Definition 15** (Laplace-Beltrami operator)**.** Given a Riemannian manifold $(\mathcal{M}, g)$, the *Laplace-Beltrami* operator $\Delta_g : \mathcal{H}^s(\mathcal{M}) \to \mathcal{H}^{s-2}(\mathcal{M})$ acts on a smooth function $f : \mathcal{M} \to \mathbb{R}$ by

$$\Delta_g f = \mathrm{div}_g(\mathrm{grad}_g(f)).$$

Moreover, $\Delta_g f$ has an equivalent weak formulation (Evans, 2022), as the unique continuous linear operator $\Delta_g : \mathcal{H}^s(\mathcal{M}) \to \mathcal{H}^{s-2}(\mathcal{M})$ which is a solution to the equation,

$$\int_{\mathcal{M}} \psi(x) \Delta_g \phi(x) d\,\mathrm{vol}_g(x) + \int_{\mathcal{M}} \langle \nabla_g \psi(x), \nabla_g \phi(x) \rangle_g d\,\mathrm{vol}_g(x) = 0, \forall \phi, \psi \in \mathcal{H}^s(\mathcal{M}). \tag{2}$$

The Laplace-Beltrami operator $\Delta_g$ is self-adjoint, eliptic and diagonalizable in $L^p(\mathcal{M})$ (Chavel, 1984; Evans, 2022), yielding a sequence of orthonormal eigenfunctions $\phi_{\lambda, \ell} \in L^2(\mathcal{M})$, where $\lambda \in \{\lambda_0, \lambda_1, \ldots\} \subseteq [0, \infty)$ represents the eigenvalue corresponding to the eigenfunction $\phi_{\lambda, \ell}$, and $\ell \in [m_\lambda]$ indexes the multiplicity of $\lambda$, denoted by $m_\lambda$ such that $\Delta_g \phi_{\lambda_i, \ell} + \lambda_\ell \phi_{\lambda_i, \ell} = 0$ for all $\ell \in \{1, \ldots, m_{\lambda_i}\}$. Note that the basis starts with the constant function $\phi_0 \equiv 1$ and $\lambda_0 = 0$. Hence, one can write $\Delta_g f = -\sum_{i=0}^{\infty} \sum_{\ell=1}^{m_{\lambda_i}} \lambda_i \langle f, \phi_{\lambda_i, \ell} \rangle \phi_{\lambda_i, \ell}$.

**Lemma 2.** *For any function $f \in L^2(\mathcal{M})$, such that $f$ is decomposed into the basis $\{\phi_{\lambda, \ell}\}_{\lambda=1}^{\infty}$ as $f = \sum_{i=0}^{\infty} \sum_{\ell=1}^{m_{\lambda_i}} \langle f, \phi_{\lambda_i, \ell} \rangle_{L^2(\mathcal{M})} \phi_{\lambda, \ell}$, we know that*

$$\|\nabla_g f\|_{L^2(\mathcal{M})}^2 = \sum_{i=0}^{\infty} \sum_{\ell=1}^{m_{\lambda_i}} \lambda_i \langle f, \phi_{\lambda_i, \ell} \rangle_{L^2(\mathcal{M})}^2,$$

*for convergent summations.*

*Proof.* By Equation (2),

$$\|\nabla_g f\|_{L^2(\mathcal{M})}^2 = \int_{\mathcal{M}} \langle \nabla_g f(x), \nabla_g f(x) \rangle_g d\,\mathrm{vol}_g(x)$$

$$= -\int_{\mathcal{M}} f(x) \Delta_g f(x) d\,\mathrm{vol}_g(x)$$

$$= \sum_{i=0}^{\infty} \sum_{\ell=1}^{m_{\lambda_i}} \lambda_i \langle f, \phi_{\lambda_i, \ell} \rangle_{L^2(\mathcal{M})}^2.$$

$\square$

## A.5 Commutativity of Laplacian and Isometric Group Actions

Let $G$ be a group acting isometrically on a compact, smooth, boundaryless manifold $\mathcal{M}$. As we stated in the main body of the paper, we have $\Delta_{\mathcal{M}}(T_g\phi) = T_g(\Delta_{\mathcal{M}}(\phi))$ for each smooth function $\phi$ on manifold $\mathcal{M}$, where $T_g\phi = \phi(gx)$. To see how, note that by Equation (2), this is equivalent to showing that

$$\int_{\mathcal{M}} h\Delta_{\mathcal{M}}(T_g\phi)d\operatorname{vol}_g(x) = \int_{\mathcal{M}} hT_g(\Delta_{\mathcal{M}}(\phi))d\operatorname{vol}_g(x), \tag{3}$$

for each smooth function $h$ on manifold $\mathcal{M}$. By changing the variables in the integrable and noting that $dx = d(gx)$ from isometry, we have

$$\int_{\mathcal{M}} h\Delta_{\mathcal{M}}(T_g\phi)d\operatorname{vol}_g(x) = -\int_{\mathcal{M}} \langle \nabla h, \nabla T_g\phi \rangle_g d\operatorname{vol}_g(x) \tag{4}$$

$$= -\int_{\mathcal{M}} \langle \nabla T_{g^{-1}}h, \nabla \phi \rangle_g d\operatorname{vol}_g(x) \tag{5}$$

$$= \int_{\mathcal{M}} T_{g^{-1}}h\Delta_{\mathcal{M}}(\phi)d\operatorname{vol}_g(x) \tag{6}$$

$$= \int_{\mathcal{M}} hT_g(\Delta_{\mathcal{M}}(\phi))d\operatorname{vol}_g(x). \tag{7}$$

## A.6 Weyl's Law under Invariances

Weyl's law characterizes the asymptotic distribution of the eigenvalues in a closed-form formula (Hörmander, 1968; Sogge, 1988; Canzani, 2013). Let us denote dimension of the space spanned by the eigenvectors corresponding to eigenvalue of the Laplace-Beltrami operator up to $\lambda$ as

$$D_\lambda := \sum_{\lambda' \le \lambda} m_{\lambda'}.$$

**Theorem 3** (Weyl's law (Hörmander, 1968; Sogge, 1988; Canzani, 2013)). *Let $(\mathcal{M}, g)$ be a compact, boundaryless $d$-dimensional Riemannian manifold. The asymptotic behavior of dimension count $D_\lambda$ follows*

$$D_\lambda = \frac{\omega_d \operatorname{vol}(\mathcal{M})}{(2\pi)^d}\lambda^{d/2} + \mathcal{O}(\lambda^{(d-1)/2}),$$

*where $\omega_d = \frac{\pi^{d/2}}{\Gamma(\frac{d}{2}+1)}$ is the volume of the unit $d$-dimensional ball in $\mathbb{R}^d$, $vol(\mathcal{M})$ is the Riemannian volume of $\mathcal{M}$, and $\mathcal{O}(\lambda^{(d-1)/2})$ represents the error term.*

Define $D_{\lambda,G}$ as the dimension of the space induced by projection of the corresponding eigenspaces of $D_\lambda$ into the space of $G$-invariant functions. Tahmasebi & Jegelka (2023) proved the following characterization over this dimension as $\lambda \to \infty$.

**Theorem 4** (Dimension counting (Tahmasebi & Jegelka, 2023)). *Let $(\mathcal{M}, g)$ be a compact, boundaryless $d$-dimensional Riemannian manifold, and $G$ be a compact finite Lie group acting isometrically on $(\mathcal{M}, g)$. Then.*

$$D_{\lambda,G} = \frac{\omega_d \operatorname{vol}(\mathcal{M}/G)}{(2\pi)^d}\lambda^{d/2} + \mathcal{O}(\lambda^{(d-1)/2}),$$

*as $\lambda \to \infty$, where again $\omega_d$ is the volume of the unit $d$-dimensional ball in $\mathbb{R}^d$.*

## A.7 Sobolev Spaces on Manifolds

The ordinary definition of Sobolev spaces on manifolds deals with having square-integrable derivatives up to an order $s$. Here, since our focus is on the spectral approach, we present the spectral definition of Sobolev spaces.

**Definition 16** (Sobolev spaces)**.** The Sobolev space of functions $H^s(\mathcal{M})$ on a compact, smooth, boundaryless Riemannian manifold $\mathcal{M}$ is defined as:

$$H^s(\mathcal{M}) \coloneqq \left\{ f = \sum_{\lambda} \sum_{\ell=1}^{m_\lambda} f_{\lambda,\ell} \phi_{\lambda,\ell}(x) : \|f\|_{H^s(\mathcal{M})}^2 \coloneqq \sum_{\lambda} \sum_{\ell=1}^{m_\lambda} D_\lambda^\alpha f_{\lambda,\ell}^2 < \infty \right\},$$

where $\alpha \coloneqq 2s/d$.

Note that the above definition is equivalent to the other definition of the Sobolev spaces that involves considering $\lambda^s$ instead of $D_\lambda^\alpha$ above. Using Weyl's law (see Appendix A.6), one can show that both definitions are equivalent.

### A.8 Sobolev Kernels

Sobolev spaces are RKHS when $s > d/2$. Indeed, the Sobolev kernel can be defined as:

$$K_{H^s(\mathcal{M})}(x,y) \coloneqq \sum_{\lambda} \sum_{\ell=1}^{m_\lambda} D_\lambda^{-\alpha} \phi_{\lambda,\ell}(x) \phi_{\lambda,\ell}(y).$$

Note that any group $G$ that acts isometrically on the manifold, also acts on the eigenspace of Laplacian via orthogonal matrices. Since orthogonal matrices preserve the inner product we conclude that

$$K_{H^s(\mathcal{M})}(gx, gy) = K_{H^s(\mathcal{M})}(x,y),$$

for any $g \in G$, which means that the Sobolev kernel is shift-invariant. However, this is clearly not $G$-invariant since it produces small bump functions, which need not be invariant.

### A.9 Kernel Ridge Regresion (KRR)

Consider a Positive-Definite Symmetric (PDS) kernel $K(.,.)$ on a smooth, compact, boundaryless manifold with $H$ denoting its RKHS. The objective of Kernel Ridge Regression (KRR) estimator is to introduce the RKHS norm to the ERM objective to make sure of finding smooth interpolators:

$$\min_{f \in H} \left\{ \frac{1}{n} \sum_{i=1}^{n} (f(x_i) - y_i)^2 + \eta \|f\|_H^2 \right\}, \tag{8}$$

where $\eta$ denotes the regularization parameter that balances the bias and variance terms. Here, the objective function takes a closed-form solution to the represented theorem for kernels. This gives an efficient estimator, which is termed KRR in the literature.

However, this estimator need not be $G$-invariant even when trained on invariant data. To see why, note that as long as the space $H$ includes non-invariant functions, there is a chances that we find a non-invariant function optimizing the above objective due to the observation noise. Thus, the only way to make sure that the KRR estimator is $G$-invariant is to impose the assumption of having $G$-invariant kernels, which translated to group averaging over the Sobolev kernel:

$$K_{H^s(\mathcal{M})}^G(x,y) \coloneqq \frac{1}{|G|} \sum_{g \in G} K_{H^s(\mathcal{M})}(gx, y). \tag{9}$$

This method is unfortunately not computationally feasible, even though it achieves minimax optimal generalization bounds for learning under invariances with kernels (Tahmasebi & Jegelka, 2023).

## B Proofs

### B.1 Minimal generating set of a Group

Here, we restate and prove the following lemma on the size of the minimal generating set in group theory for completeness.

**Proposition 5.** *The minimal generating set $S$ of a finite group $G$, has a size $\rho(G) \coloneqq |S| \leq \log_2(|G|)$.*

*Proof.* Consider the minimal generating set $S = \{g_1, g_2, \ldots, g_{|S|}\}$ of the finite group $G$. For each $k \in \{1, 2, \ldots, |S|\}$, define $G_k = \langle g_1, g_2, \ldots, g_k \rangle$.

The identity $e$ is not equal to any of $g_k$, and hence cannot be a member of any $G_n$, since it can always be produced by combining an element with its inverse. Moreover, for all $k \in \{1, 2, \ldots, |S|\}$, we know that $g_{k+1} \notin G_k$, since otherwise $\langle g_1, g_2, \ldots, g_k, g_{k+2}, \ldots g_{|S|} \rangle = G$ which contradicts the minimality of the generating set $S$ for the group $G$. Therefore, $g_{n+1} G_n$ the left coset of $G_n$ is disjoint from $G_n$. Additionally, by definition, we know that $g_{n+1} G_n \bigcup G_n \subseteq G_{n+1}$. Hence, $|G_{n+1}| \geq |g_{n+1} G_n| + |G_n| = 2|G_n|$. By induction, $2^{|S|} = 2^{|S|}|G_1| \leq |G_{|S|}| = |G|$ which establishes the claim. $\square$

### B.2 CONSTRAINED OPTIMIZATION

In this section, we preset a detailed analysis of the constrained quadratic optimization problem that is used in Algorithm 1.

**Proposition 6** (Projection into invariant subspace of eigenspaces)**.** *The optimization problem,*

$$\widehat{f}_\lambda := \arg\min_{f_\lambda} \sum_{\ell=1}^{m_\lambda} (f_{\lambda,\ell} - \widetilde{f}_{\lambda,\ell})^2, \tag{10}$$

$$s.t. \quad \forall g \in S : D^\lambda(g) f_\lambda = f_\lambda,$$

*with $|S| = m$, has a closed form solution,*

$$\widehat{f}_\lambda = \widetilde{f}_\lambda - B^{\lambda\top}(B^\lambda B^{\lambda\top})^\dagger (B^\lambda \widetilde{f}_\lambda),$$

*where*

$$B^\lambda = \begin{bmatrix} D^\lambda(g_1) - I \\ D^\lambda(g_2) - I \\ \vdots \\ D^\lambda(g_m) - I \end{bmatrix},$$

*and $\dagger$ denotes Moore–Penrose inverse.*

*Proof.* For better readability, we define $B(g_i) := D^\lambda(g_i) - I$, where $I \in \mathbb{R}^{m_\lambda \times m_\lambda}$ is the identity matrix of size $m_\lambda$, then,

$$B^\lambda = \begin{bmatrix} B(g_1) \\ B(g_2) \\ \vdots \\ B(g_m) \end{bmatrix}.$$

For ease of notation, let $a := \widetilde{f}_\lambda \in \mathbb{R}^{m_\lambda}$ and $a^* := \widehat{f}_\lambda \in \mathbb{R}^{m_\lambda}$, then the optimization problem (10) can be rewritten as,

$$a^* = \min_{a'} \frac{1}{2}\|a - a'\|^2 \text{ subject to } B^\lambda a' = 0. \tag{11}$$

Now, we need to show that the projection of $a$ onto the subspace defined by $\{a' \mid Ba' = 0\}$ has the following analytical form,

$$a^* = a - B^{\lambda\top}(B^\lambda B^{\lambda\top})^\dagger(B^\lambda a).$$

We form the Lagrangian,

$$\mathcal{L}(a', \lambda) = \frac{1}{2}\|a - a'\|^2 + \xi^\top B^\lambda a,$$

where $\xi \in \mathbb{R}^{m_\lambda}$ is the vector of Lagrange multipliers. By taking gradients,

$$\frac{\partial \mathcal{L}}{\partial a'} = (a' - a) + B^{\lambda\top}\xi = 0,$$

thus,

$$a^* = a - {B^\lambda}^\top \xi. \tag{12}$$

Substituting back into the constraint $B^\lambda a^* = 0$,

$$B^\lambda (a - {B^\lambda}^\top \xi) = B^\lambda a - B^\lambda {B^\lambda}^\top \xi = 0. \tag{13}$$

Hence, $\xi$ satisfies the above linear system which may have infinite number of solutions. We claim that the choice of $\xi^* = (B^\lambda {B^\lambda}^\top)^\dagger B^\lambda a$ leads to the optimal solution of optimization problem (11). The objective of optimization (11) is $\|a - a^*\|^2 = \|{B^\lambda}^\top \xi\|_2^2$. Any solution $\xi$ to the linear system (13), can be decomposed as $\xi = \xi^* + \xi_0$ where $\xi_0$ is in the nullspace of $B^\lambda {B^\lambda}^\top$. Hence,

$$\|{B^\lambda}^\top \xi\|_2^2 = \|{B^\lambda}^\top (\xi^* + \xi_0)\|_2^2 \overset{(I)}{=} \|{B^\lambda}^\top \xi^*\|_2^2 + \|{B^\lambda}^\top \xi_0\|_2^2 \geq \|{B^\lambda}^\top \xi^*\|_2^2.$$

(I) follows since ${B^\lambda}^\top \xi^*$ and ${B^\lambda}^\top \xi_0$ are orthogonal w.r.t. each other. Placing $\xi^* = (B^\lambda {B^\lambda}^\top)^\dagger B^\lambda a$ in Equation (12) concludes the proof. □

*Remark* 7 (Time complexity of optimization in each eigenspace). We arbitrarily chose to use the closed-form solution of the optimization problem (10) instead of iterative approaches. In the closed form solution, we need to calculate the pseudoinverse of matrix $B^\lambda {B^\lambda}^\top \in \mathbb{R}^{|S|m_\lambda \times |S|m_\lambda}$ which can be done via singular value decomposition (SVD) in $O(|S|^3 m_\lambda^3)$. The other operations are matrix multiplications that are dominated by this part in terms of computational complexity.

### B.3 MAIN THEOREM

**Theorem 1** (Learning with exact invariances in polynomial time). *Consider the problem of learning with invariances with respect to a finite group $G$ using a labeled dataset of size $n$ sampled from a manifold of dimension $d$. Assume that the optimal regression function belongs to the Sobolev space of functions of order $s$, i.e., $f^\star \in H^s(\mathcal{M})$ for some $s > d/2$ and let $\alpha := 2s/d$.* Then, there exists an algorithm that, given the data, produces an exactly invariant estimator $\widehat{f}$ such that:

- *It operates in time $\mathcal{O}\big(\log^3(|G|)n^{3/(1+\alpha)} + n^{(2+\alpha)/(1+\alpha)}\big)$;*

- *It achieves an excess population risk (or generalization error) of $\mathcal{R}(\widehat{f}) = \mathcal{O}\big(n^{-s/(s+d/2)}\big)$;*

- *It requires $\mathcal{O}\big(\log(|G|)n^{2/(1+\alpha)} + n^{(2+\alpha)/(1+\alpha)}\big)$ oracle calls to construct the estimator;*

- *For any $x \in \mathcal{M}$, the estimated label $\widehat{f}(x)$ can be computed in time $\mathcal{O}\big(n^{1/(1+\alpha)}\big)$ using $\mathcal{O}\big(n^{1/(1+\alpha)}\big)$ oracle calls.*

*Proof.* To prove Theorem 1, we use Algorithm 1. Let us start by calculating the time and oracle complexity of the algorithm. Given a dataset $\mathcal{S}$ of size $n$, we first compute

$$\widetilde{f}_{\lambda,\ell} = \frac{1}{n} \sum_{i=1}^n y_i \phi_{\lambda,\ell}(x_i), \tag{14}$$

for each $\lambda$ such that $D_\lambda \leq D = n^{1/(1+\alpha)}$, and each $\ell \in [m_\lambda]$. This requires $\mathcal{O}(n^{1+1/(1+\alpha)})$ oracle calls and can be accomplished in time $\mathcal{O}(n^{1+1/(1+\alpha)})$.

Next, we solve the following constrained quadratic program:

$$\widehat{f}_{\lambda,\ell} \leftarrow \arg\min_{f_{\lambda,\ell}} \sum_{\ell=1}^{m_\lambda} (f_{\lambda,\ell} - \widetilde{f}_{\lambda,\ell})^2, \tag{15}$$

$$\text{s.t.} \quad \forall g \in S : D^\lambda(g) f_\lambda = f_\lambda. \tag{16}$$

This is done for each $\lambda$ such that $D_\lambda \leq n^{1/(1+\alpha)}$. Note that to even set up this program, we need $\mathcal{O}(|S|m_\lambda^2)$ oracle calls to find the constraints. We have

$$(D^\lambda(g))_{\ell,\ell'} = \langle \phi_{\lambda,\ell}(x), \phi_{\lambda,\ell'}(gx) \rangle_{L^2(\mathcal{M})} \tag{17}$$

for each $\ell, \ell' \in [m_\lambda]$.

Therefore, the total oracle complexity of the proposed algorithm is

$$\mathcal{O}\left(\sum_{\lambda:D_\lambda \leq n^{1/(1+\alpha)}} |S|m_\lambda^2 + n^{(2+\alpha)/(1+\alpha)}\right). \tag{18}$$

We have already shown in Proposition 5 that one can use a generator set with $\rho(G) \leq \log(|G|)$. Moreover, note that

$$\sum_{\lambda:D_\lambda \leq n^{1/(1+\alpha)}} m_\lambda^2 = \mathcal{O}(n^{2/(1+\alpha)}). \tag{19}$$

Therefore, the oracle complexity is

$$\mathcal{O}\left(\log(|G|)n^{2/(1+\alpha)} + n^{(2+\alpha)/(1+\alpha)}\right). \tag{20}$$

Let us now calculate the time complexity of finding the estimator. We have already established that we can compute the empirical estimation in time $\mathcal{O}(n^{1+1/(1+\alpha)})$. Next, we need to solve the constrained quadratic program with $\log(|G|)$ constraints and $m_\lambda$ variables for each $\lambda$ such that $D_\lambda \leq n^{1/(1+\alpha)}$. Using the proposed algorithm in Appendix B.2 and also Remark 7, we can solve each of these constrained quadratic programs in time $\mathcal{O}(\log^3(|G|)m_\lambda^3)$. Therefore, the total time complexity of this step is bounded by

$$\mathcal{O}\left(\sum_{\lambda:D_\lambda \leq n^{1/(1+\alpha)}} \log^3(|G|)m_\lambda^3\right) = \mathcal{O}\left(\log^3(|G|)n^{3/(1+\alpha)}\right). \tag{21}$$

This proves that the total time complexity of Algorithm 1 is

$$\mathcal{O}\left(\log^3(|G|)n^{3/(1+\alpha)} + n^{(2+\alpha)/(1+\alpha)}\right). \tag{22}$$

Finally, note that given $\widehat{f}$, one can evaluate it on new unlabeled data $x \in \mathcal{M}$ using the formula:

$$\widehat{f}(x) = \sum_{\lambda:D_\lambda \leq D} \sum_{\ell=1}^{m_\lambda} \widehat{f}_{\lambda,\ell}\phi_{\lambda,\ell}(x), \tag{23}$$

with $D = n^{1/(1+\alpha)}$, which requires both time and oracle complexity of $\mathcal{O}(n^{1/(1+\alpha)})$.

To complete the proof, we need to study the convergence of the population risk of the proposed estimator. We first note that

$$\mathcal{R}(\widehat{f}) = \mathbb{E}[\|\widehat{f} - f^\star\|_{L^2(\mathcal{M})}^2] \leq 2\mathbb{E}[\|\widehat{f} - f^\star_{\leq D}\|_{L^2(\mathcal{M})}^2] + 2\mathbb{E}[\|f^\star_{>D}\|_{L^2(\mathcal{M})}^2], \tag{24}$$

where $f^\star_{\leq D}$ denotes the orthogonal projection of the function $f^\star$ onto the space of eigenfunctions with eigenvalues satisfying $D_\lambda \leq D$. Moreover, $f^\star_{>D} = f^\star - f^\star_{\leq D}$.

First, let us upper bound the second term. Note that, according to the assumption, $f^\star \in H^s(\mathcal{M})$. Thus,

$$\mathbb{E}[\|f^\star_{>D}\|_{L^2(\mathcal{M})}^2] = \sum_{\lambda:D_\lambda > D} \sum_{\ell=1}^{m_\lambda} (f^\star_{\lambda,\ell})^2 \tag{25}$$

$$= \sum_{\lambda:D_\lambda > D} \sum_{\ell=1}^{m_\lambda} D_\lambda^{-\alpha} D_\lambda^\alpha (f^\star_{\lambda,\ell})^2 \tag{26}$$

$$\leq D^{-\alpha} \sum_{\lambda:D_\lambda > D} \sum_{\ell=1}^{m_\lambda} D_\lambda^\alpha (f^\star_{\lambda,\ell})^2 \tag{27}$$

$$\leq D^{-\alpha} \sum_{\lambda} \sum_{\ell=1}^{m_\lambda} D_\lambda^\alpha (f^\star_{\lambda,\ell})^2 \tag{28}$$

$$= D^{-\alpha} \|f^\star\|_{H^s(\mathcal{M})}^2. \tag{29}$$

Now we focus on the first term. Note that

$$\mathbb{E}[\|\widehat{f} - f^\star_{\leq D}\|^2_{L^2(\mathcal{M})}] = \sum_{\lambda:D_\lambda \leq D} \sum_{\ell=1}^{m_\lambda} \mathbb{E}[|\widehat{f}_{\lambda,\ell} - f^\star_{\lambda,\ell}|^2]. \tag{30}$$

According to the definition, we have

$$f^\star_{\lambda,\ell} = \mathbb{E}_x[f^\star(x)\phi_{\lambda,\ell}(x)] = \mathbb{E}_{x,y}[y\phi_{\lambda,\ell}(x)], \tag{31}$$

for each $\lambda, \ell$. Moreover, $\widetilde{f}_{\lambda,\ell}$ is the empirical estimation obtained from data:

$$\widetilde{f}_{\lambda,\ell} = \frac{1}{n} \sum_{i=1}^{n} y_i \phi_{\lambda,\ell}(x_i). \tag{32}$$

Thus, we obtain

$$\mathbb{E}[|\widetilde{f}_{\lambda,\ell} - f^\star_{\lambda,\ell}|^2] = \frac{1}{n} \mathbb{E}\left[|y\phi_{\lambda,\ell}(x) - \mathbb{E}[y\phi_{\lambda,\ell}(x)]|^2\right] \tag{33}$$

$$= \frac{1}{n} \mathbb{E}\left[|\epsilon\phi_{\lambda,\ell}(x) + f^\star(x)\phi_{\lambda,\ell}(x) - \mathbb{E}[f^\star(x)\phi_{\lambda,\ell}(x)]|^2\right] \tag{34}$$

$$= \frac{1}{n}\left(\sigma^2 \mathbb{E}[\phi^2_{\lambda,\ell}] + \mathbb{E}\left[|f^\star(x)\phi_{\lambda,\ell}(x) - \mathbb{E}[f^\star(x)\phi_{\lambda,\ell}(x)]|^2\right]\right) \tag{35}$$

$$\leq \frac{1}{n}\left(\sigma^2 + \mathbb{E}[f^\star(x)^2\phi^2_{\lambda,\ell}(x)]\right) \tag{36}$$

$$\leq \frac{1}{n}\left(\sigma^2 + \|f^\star\|^2_{L^\infty(\mathcal{M})}\right), \tag{37}$$

where we used the orthonormality of the eigenfunctions $\phi_{\lambda,\ell}$. Then, summing this up to dimension $D$ gives:

$$\mathbb{E}[\|\widetilde{f} - f^\star_{\leq D}\|^2_{L^2(\mathcal{M})}] \leq \frac{D}{n}\left(\sigma^2 + \|f^\star\|^2_{L^\infty(\mathcal{M})}\right). \tag{38}$$

Note that, by definition, $\widehat{f} = P_G\widetilde{f}$, where $P_G : L^2(\mathcal{X}) \to L^2(\mathcal{X})$ is the orthogonal projection operator onto the invariant functions. Therefore, we have

$$\mathbb{E}[\|\widehat{f} - f^\star_{\leq D}\|^2_{L^2(\mathcal{M})}] = \mathbb{E}[\|P_G\widetilde{f} - f^\star_{\leq D}\|^2_{L^2(\mathcal{M})}] \tag{39}$$

$$= \mathbb{E}[\|P_G\widetilde{f} - P_G f^\star_{\leq D}\|^2_{L^2(\mathcal{M})}] \tag{40}$$

$$\leq \mathbb{E}[\|\widetilde{f} - f^\star_{\leq D}\|^2_{L^2(\mathcal{M})}] \tag{41}$$

$$\leq \frac{D}{n}\left(\sigma^2 + \|f^\star\|^2_{L^\infty(\mathcal{M})}\right), \tag{42}$$

where the penultimate step follows from $P_G f^\star_{\leq D} = f^\star_{\leq D}$.

Therefore, we can combine the two terms to derive the following population risk bound:

$$\mathcal{R}(\widehat{f}) = \mathbb{E}[\|\widehat{f} - f^\star\|^2_{L^2(\mathcal{M})}] \leq \frac{D}{n}\left(\sigma^2 + \|f^\star\|^2_{L^\infty(\mathcal{M})}\right) + D^{-\alpha}\|f^\star\|^2_{H^s(\mathcal{M})}. \tag{43}$$

We can now specify the above bound to $D = n^{1/(1+\alpha)}$, which is used in the algorithm, to get:

$$\mathcal{R}(\widehat{f}) = \mathbb{E}[\|\widehat{f} - f^\star\|^2_{L^2(\mathcal{M})}] \leq n^{-\alpha/(1+\alpha)}\left(\sigma^2 + \|f^\star\|^2_{L^\infty(\mathcal{M})}\right) + n^{-\alpha/(1+\alpha)}\|f^\star\|^2_{H^s(\mathcal{M})}, \tag{44}$$

which is equivalent to

$$\mathcal{R}(\widehat{f}) = \mathcal{O}(n^{-\alpha/(1+\alpha)}). \tag{45}$$

This completes the proof.

*Remark* 8. Note that other choices of $D$ may or may not yield better bounds depending on the sparsity of the solution. For this sparsity-unaware upper bound that we use, such a choice of $D$ is optimal. Additionally, since we focus on polynomial time algorithms, we cannot choose exponentially large $D$ even if they deliver gains in sample complexity.

$\square$

## C  EXPERIMENTS

In this section, we provide complementary experiments to support our theoretical results. We first show that, in practice, Kernel Ridge Regression (KRR) is not a $G$-invariant estimator. Then, we demonstrate that our algorithm (Spec-Avg) achieves the same rate of population risk as KRR, while enjoying exact invariance properties.

### C.1  PROBLEM STATEMENT

We consider the input space (manifold) $\mathbb{T}^d = [-1, 1)^d$, which represents a flat $d$-dimensional torus. Additionally, we consider the group of sign-invariances $G = \{\pm 1\}^d$, acting on this space via coordinate-wise sign inversions. The dataset is generated as $n$ independent and identically distributed (i.i.d.) samples drawn uniformly from this space, with the target function defined as:

$$f^*(x) = \frac{1}{d} \sum_{i=1}^{d} i x_i^2.$$

Clearly, this function is invariant w.r.t. group action $G$. To analyze estimation via kernels in this setup, we consider a periodic kernel on the torus $\mathbb{T}^d$, specifically the *von Mises Kernel* (von Mises, 1918; Mardia & Jupp, 2009), defined as:

$$K_\eta(x, y) = \exp\left(\eta \cos(\pi(x - y))\right),$$

where $\eta$ is a positive parameter controlling the kernel's sharpness. This kernel function is particularly useful for circular and directional statistics. Moreover, the kernel admits the following sign-invariant eigenfunctions:

$$\phi_{\ell_1, \ell_2, \ldots, \ell_d}(x) = \prod_{i=1}^{d} \cos(\pi \ell_i x_i),$$

where $\ell_i \in \mathbb{N} \cup \{0\}$. The corresponding eigenvalues can be computed as

$$\lambda = \pi \sum_{i=1}^{d} \ell_i^2,$$

derived from the partial differential equation

$$\Delta \phi_{\ell_1, \ell_2, \ldots, \ell_d} + \lambda \phi_{\ell_1, \ell_2, \ldots, \ell_d} = 0.$$

This formulation facilitates the analysis of KRR and Spec-Avg under symmetry constraints, ensuring their compatibility with the underlying group structure. It is worth noting that, in this setting, $|G| = 2^d$. Consequently, methods based on group averaging are computationally inefficient due to the exponential growth of the group size with the dimensionality $d$.

### C.2  SETTINGS

We conduct our experiments for $d = 10$. The trained models are evaluated on a test dataset of size 100. Both the test and train datasets are generated uniformly from the interval $[-1, 1]^d$, independently and identically distributed. Each point in our plots represents an average over 10 different random seeds (from 1 to 10) to account for the randomness in the data generation process.

### C.3  RESULTS

The results of the experiments are depicted in Figure 1 and Figure 2.

While our algorithm (Spec-Avg) is $G$-invariant by construction, there is no theoretical guarantee for Kernel Ridge Regression (KRR) to be $G$-invariant. In Figure 1, we demonstrate that this is indeed the case in practice, as the estimator KRR is not $G$-invariant. We define the following measure of Invariance Discrepancy:

$$\mathrm{ID}(\widehat{f}) \stackrel{\text{def}}{=} \sup_{x \in \mathcal{X}, g \in G} |\widehat{f}(x) - \widehat{f}(gx)|,$$

where $\widehat{f}$ is the estimator. We report this value for KRR across different choices of the regularization parameter $\lambda$. It is worth noting that $\mathrm{ID}(\widehat{f})$ is zero for the Spec-Avg estimator, as it is $G$-invariant by design.

In Figure 2, we present the empirical excess population risk of KRR and Spec-Avg for different hyperparameters $\lambda$ and $D$, respectively. As expected, it is demonstrated that with an appropriate choice of hyperparameters, KRR and Spec-Avg achieve the same order of test error. Higher values of the regularization parameter $\lambda$ for KRR correspond to lower values of the sparsity parameter $D$ for Spec-Avg, both of which act as mechanisms for regularizing the norm of the estimator. It can be observed that Spec-Avg with $D = 176$ achieves the same order of performance as KRR with $\lambda = 50$.

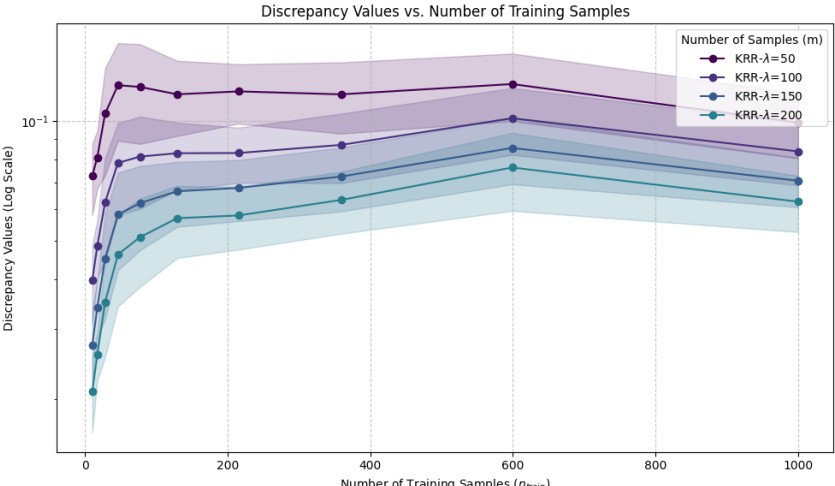

Figure 1: Invariance Discrepancy measure of Kernel Ridge Regression (KRR) for various choices of the regularization parameter $\lambda$. The resulting estimator, KRR, is not invariant with respect to the group $G$ of sign averages $\{\pm 1\}^d$, whereas Spec-Avg is $G$-invariant by construction. Each point in the plot represents an average over 10 different random seeds. The Invariance Discrepancy measure used for this plot is defined as $\sup_{x \in \mathcal{X}, g \in G} |\widehat{f}(x) - \widehat{f}(gx)|$, where $\widehat{f}$ is the estimator. The set $\mathcal{X}$ consists of 100 points uniformly sampled from the interval $[-1, 1]^d$, independently and identically distributed.

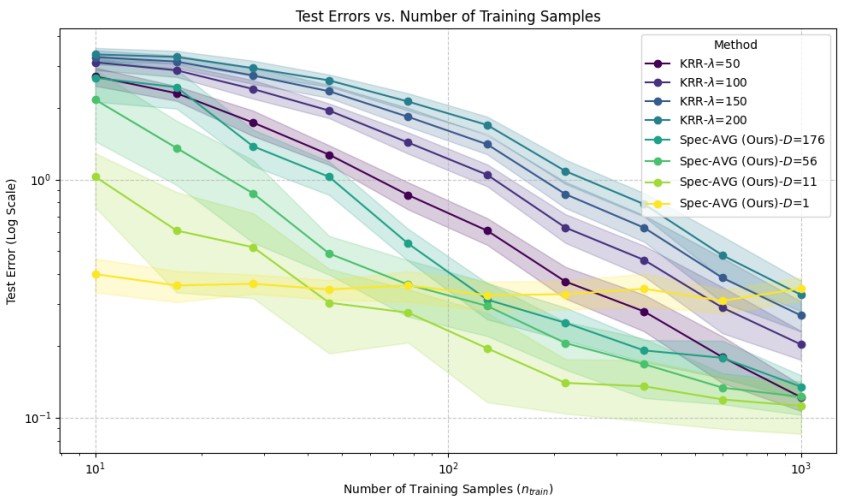

Figure 2: Test error (empirical excess population risk) of KRR for different choices of the regularization parameter $\lambda$ and Spec-Avg for different choices of the sparsity parameter $D$. Conceptually, higher values of $\lambda$ and lower values of $D$ encourage sparser representations for the estimators KRR and Spec-Avg, respectively. As suggested by our theory, it can be observed that test error rates of the same order can be achieved by Spec-Avg and KRR with appropriate choices of hyperparameters. Note that the test errors are shown on a log scale. Their almost linear behavior implies that they are polynomial functions of the number of training samples with comparable orders. We note that each point in the plot represents an average over 10 different random seeds.

