# OpenReview forum: "Learning with Exact Invariances in Polynomial Time"
_ICLR.cc/2025/Conference — Submitted to ICLR 2025_

### Official Review · Reviewer_LDcF · 2024-10-29

**Soundness:** 3
**Presentation:** 2
**Contribution:** 3
**Rating:** 6
**Confidence:** 4

**Summary:**

This paper studied the statistical-computational trade-offs for learning with exact invariances using kernel regression over manifolds,
proposed a polynomial-time algorithtm for the first time and proved the generalization error bound of the algorithm
by utilizing some tools from differential geometry, spectral theory, and optimization.

**Strengths:**

Learning with invariance (or other properties of learning problem) is important for both theoretical studies and algorithm design in machine learning. This paper proposes the first computationally efficient algorothm for kernel leanring over manifolds,
and introduces some new theoretical tools from differential geometry and spectral theory,
which I think might be of independent interest.

**Weaknesses:**

The main weakness of this paper is the generalization bound which is same with learning without invariance,
failing to demonstrate the benefit introduced by invariance.
Although it has been mentioned as a future work, it would be more interesting if the current algorithm already satisfies this property.
In contrast to the bound in [1],
the upper bound in this paper not only maintains a larger problem-dependent constant but also a larger dependence on the norm of $f^\ast$.

References

[1] Behrooz Tahmasebi and Stefanie Jegelka. The Exact Sample Complexity Gain from Invariances for Kernel Regression. NeurIPS, 2023.

**Questions:**

(1) In line 150, whether the expectation operator $\mathbb{E}_{S}$ should be corrected as $\mathbb{E}$.
I think the expectation is w.r.t. the distribution over manifolds not the trainning data.

(2) The current analysis of the generalization error just used standard techniques, but did not make full use of the invariance.
	Is the reason that the current analysis is not tight for the proposed algorithm, or that the proposed algorithm did not fully leverage the invariance?

---

> ### Author Response · Authors · 2024-11-21
> **Response to Reviewer LDcF**
>
> We express our gratitude to the reviewer for their valuable comments and feedback. In the above, we have offered a general response to the reviewers, addressing some of their common concerns about experiments. Here, we provide additional responses to the remaining questions raised.
>
> > The main weakness of this paper is the generalization bound which is same with learning without invariance, failing to demonstrate the benefit introduced by invariance. Although it has been mentioned as a future work, it would be more interesting if the current algorithm already satisfies this property. In contrast to the bound in [1], the upper bound in this paper not only maintains a larger problem-dependent constant but also a larger dependence on the norm of $f^*$.
>
> Thank you very much for mentioning this point. Indeed, it is currently unknown whether the minimax optimal bound of [1] can be achieved by an estimator with polynomial-time computation. We note that, despite the differences in constant terms between our bounds and those in [1], the orders are the same. Furthermore, our bounds match the sample complexity of kernel regression without invariances.
>
> To conclude, we reiterate that the kernel estimator using group averaging [1] is minimax optimal but requires potentially super-exponential time complexity.
>
>
> > In line 150, whether the expectation operator $\mathbb{E}_S$ should be corrected as $\mathbb{E}$. I think the expectation is w.r.t. the distribution over manifolds not the trainning data.
>
> Thank you for your comment. As you correctly pointed out, we intended to indicate that the expectation is taken over uniformly random points generated from the manifold, not the empirical measure. To align with standard notation and incorporate your suggestion, we have updated our notation in the revised version. Furthermore, we have clarified the definition of the expectation involved in the population risk on Line 150 (highlighted in red) to ensure clarity.
>
>
> > The current analysis of the generalization error just used standard techniques, but did not make full use of the invariance. Is the reason that the current analysis is not tight for the proposed algorithm, or that the proposed algorithm did not fully leverage the invariance?
>
>
> Thank you for asking this question. We note that the kernel estimator using group averaging [1] is minimax optimal but requires potentially super-exponential time complexity. In contrast, we trade off this statistical optimality by introducing a new algorithm that achieves polynomial time complexity while remaining statistically desirable (though not minimax optimal).
>
> To address the reviewer's concern regarding the full utilization of invariance, we emphasize that our algorithm requires setting a cut-off frequency for estimations, denoted by $D$. The minimax optimal bound can be achieved by setting $D$ proportional to $|G|$, but this compromises the time efficiency of the proposed algorithm. To fully leverage invariance while maintaining polynomial time efficiency, one must optimize the hyperparameter $D$ such that it is at most a polynomial in $n$ and $d$, subject to non-asymptotic bounds on the low-dimensional spectral sparsity of the group action on the manifold. However, these bounds are not well understood in the applied mathematics literature for arbitrary manifolds, making the problem of optimizing $D$ under these constraints both unclear and difficult to address. As a result, we set $D$ to a predefined (potentially suboptimal) value that achieves the desired properties of our algorithm. Identifying the optimal $D$, as well as the general optimal statistical-computational trade-off, is left for future work.
>
> Finally, we note that, by construction, our algorithm outputs an **exact invariant** estimator.
>
> [1] Behrooz Tahmasebi and Stefanie Jegelka. The Exact Sample Complexity Gain from Invariances for Kernel Regression. NeurIPS, 2023.

---

> > ### Comment · Reviewer_LDcF · 2024-11-22
> >
> > Thanks for the detailed responses. The explanation on the hardness of this problem has addressed most of my concerns. I disagree that the order of the bound in this paper is the same as that in [1] because the definitions of $d$ are different in the two bounds.
> >
> > I maintain the score and recommend accepting this paper based on its contributions to both algorithm and theory.
> >
> > [1] Behrooz Tahmasebi and Stefanie Jegelka. The Exact Sample Complexity Gain from Invariances for Kernel Regression. NeurIPS, 2023.

---

> > > ### Author Response · Authors · 2024-12-01
> > > **Additional concerns?**
> > >
> > > Thank you very much for your reply. We are wondering if there are any additional concerns we should address that might lead to a more positive evaluation of our work in your view.

---

### Official Review · Reviewer_GhGN · 2024-10-31

**Soundness:** 3
**Presentation:** 3
**Contribution:** 3
**Rating:** 6
**Confidence:** 4

**Summary:**

The authors consider the setting of invariant regression when the regression function is smooth $s$-Sobolev space. It is well known that for $s>d/2$, the $s$-Sobolev space is an RKHS and for regression without invariance constraint, the ridge estimator is minimax optimal.  As with any kernel method, a forward pass through such an estimator requires at worst $n^3$ time. However, the ridge estimator may not be invariant under the group of interest. In this paper, the authors provide a new estimator that achieves the same error rate as the usual ridge estimator, is invariant under the group of interest, and can be implemented in polynomial time. Note that this estimator may not be minimax optimal as the true regression function with the required constraint lies in a smaller class.

**Strengths:**

1. The paper is well-written and is easy to follow.
2. The estimator is pretty natural. While the proof of Theorem 1 is relatively straightforward, I find the idea of re-representing the invariance constraints in the coefficients along basis expansion clever.

**Weaknesses:**

1. I did not find any apparent weakness. The paper delivers on its promise of developing an invariant estimator for Sobolev regression.

2. One might argue that the lack of experiments takes away from the paper, but I am fine with a theory paper with sound and rigorous proof not including experiments. The paper’s conceptual contribution is, in my view, sufficient to address this limitation.

**Questions:**

In Section 3, the authors mention that group averaging can be too costly for large groups. But what about group averaging only over the generator? Does the kernel ridge regression with the group averaging over the generator set $S$ yield an invariant estimator? If so, what do we know about its generalization error?

---

> ### Author Response · Authors · 2024-11-21
> **Response to Reviewer GhGN**
>
> We sincerely appreciate the reviewer's thoughtful and constructive comments. In our general response, we address some common concerns regarding the experiments. Here, we provide additional responses to the remaining specific concerns.
>
> > I did not find any apparent weakness. The paper delivers on its promise of developing an invariant estimator for Sobolev regression.
>
> Thank you very much for your positive feedback on the results!
>
> > One might argue that the lack of experiments takes away from the paper, but I am fine with a theory paper with sound and rigorous proof not including experiments. The paper’s conceptual contribution is, in my view, sufficient to address this limitation.
>
> Thank you for your comment. Given the reviewers' suggestions to evaluate the proposed algorithm through experiments, we decided to include experiments in the revised version of the paper. Please refer to our general response for details on the experiments.
>
>
> > In Section 3, the authors mention that group averaging can be too costly for large groups. But what about group averaging only over the generator? Does the kernel ridge regression with the group averaging over the generator set yield an invariant estimator? If so, what do we know about its generalization error?
>
> Thanks for your valuable comment. We note that averaging over a generator set does not yield an invariant estimator. For example, let us consider the case of linear kernel $K(x,y) = x^Ty$ along with the permutation group $P_d$ acting via permuting the coordinates of vectors. Note that $\sigma_1 = (1 \mapsto 2, 2 \mapsto 1)$ and
> $\sigma_2 = (1 \mapsto 2 \mapsto 3 \mapsto 4 \dots \mapsto d \mapsto 1)$ are two permutations generating $P_d$, and if we average the kernel over these permutations it yields $\tilde{K}(x,y) = \dfrac{1}{2}K(x, \sigma_1 y) + \dfrac{1}{2}K(x, \sigma_2 y) = x^T\tilde{y}$, where $\tilde{y} \in \mathbb{R}^d$ is defined as:
>
> $$
> \tilde{y}_{i} = \begin{cases}
>     \dfrac{1}{2}(y_2 + y_d) & \text{if } i = 1, \\\\
>     y_1 & \text{if } i = 2, \\\\
>     \dfrac{1}{2}(y_i + y\_{i-1}) & \text{if } i > 2 \text{ and } i \leq d.
> \end{cases}
> $$
>
> Clearly, the resulting kernel is not permutation-invariant and, therefore, does not necessarily produce invariant estimators (it is worth noting that it is not even a PSD kernel). Consequently, generating sets are not useful for simplifying the group averaging process involved in constructing invariant kernels.
>
> Once again, we thank the reviewer for raising this question, and we hope this example addresses their concern.

---

> > ### Comment · Reviewer_GhGN · 2024-11-22
> >
> > I want to thank the authors for answering my question. I will maintain my score.

---

> > > ### Author Response · Authors · 2024-12-01
> > > **Additional concerns?**
> > >
> > > Thank you very much for your reply. We are wondering if there are any additional concerns we should address that might lead to a more positive evaluation of our work in your view.

---

### Official Review · Reviewer_S5px · 2024-11-04

**Soundness:** 4
**Presentation:** 4
**Contribution:** 4
**Rating:** 8
**Confidence:** 4

**Summary:**

The authors show that it is possible to learn with invariance on compact smooth manifolds with a finite group action in a manner that simultaneously enjoys polynomial sample complexity and computational complexity in the problem parameters.

**Strengths:**

The paper is well-written and flows well. The exposition is very clear. The authors also highlighted the novelty and the gap that is filled. The proof sketch is nice and I find that conveys the essential technical aspects.

**Weaknesses:**

I think the first bullet point on Line 085 could be made more clear: the main results of this paper shows one point on the statistical-computational trade-off curve, as I understand. I don't see how to interpret the results as showing how to trade-off statistical efficiency to gain computational-efficiency. I might be misunderstanding this bullet point however.

**Questions:**

Is the case of (positive dimensional) lie group actions addressable using this approach?

Theorem 1 should make it clear that $\hat{f}$ is $G$-invariant.

Line 234: how about the second oracle (the one about computing inner product between shifted eigenfunctions). Is that also efficiently computable?

Are there other examples besides the sphere for which the oracle calls can be computed efficiently? E.g., if the authors can talk about an example with Stiefel manifolds, that would be nice to know about.

---

> ### Author Response · Authors · 2024-11-21
> **Response to Reviewer S5px**
>
> We sincerely appreciate the reviewer's recognition of the significance of the contributions made by our work, as well as the constructive comments provided. Please let us know if there are any other concerns that need to be addressed.
>
> > I think the first bullet point on Line 085 could be made more clear: the main results of this paper shows one point on the statistical-computational trade-off curve, as I understand. I don't see how to interpret the results as showing how to trade-off statistical efficiency to gain computational-efficiency. I might be misunderstanding this bullet point however.
>
>
>
> Thank you for your valuable comment! Here, we trade off computational complexity against statistical complexity using the parameter $D$, which represents the number of eigenspaces utilized in the estimator (equivalent to the number of convex quadratic programs solved). A larger $D$ improves statistical efficiency but increases computational cost.
>
> For the primary goal of this paper, i.e., achieving an efficient invariant estimator in polynomial time, selecting $D = n^{1/(1 + \alpha)}$ is sufficient. Please refer to Remark 8 in the manuscript for a short discussion.
>
> We have added a highlighted line (Line 91) to further clarify this point.
>
>
> > Is the case of (positive dimensional) lie group actions addressable using this approach?
>
> Thank you for your question. In this version of the paper, our primary focus is on finite groups, where group averaging can be computed in finite (though potentially prohibitively large) time. We have left the extension to positive-dimensional Lie groups as a direction for future work.
>
> It is important to note that for infinite groups, proposing an algorithm that computes group averaging in finite time is not immediately feasible (and currently unknown) due to the infinite number of group transformations involved in the averaging process.
>
>
> > Theorem 1 should make it clear that $\hat{f}$ is $G$-invariant.
>
>
> Thanks for mentioning this! We made necessery changed in Line 249 in the revised version (highlighted in red).
>
> > Line 234: how about the second oracle (the one about computing inner product between shifted eigenfunctions). Is that also efficiently computable?
>
> Yes, for the case of (harmonic) polynomials, as long as they have bounded degree (Line 233), one can compute their inner product using polynomial multiplication algorithms (even naive algorithms run in polynomial time for this problem, though faster ones, such as those based on the Fast Fourier Transform, exist in the literature). Therefore, both oracles can be answered in polynomial time. We have also applied these clarifications to the revised version (Line 235, highlighted in red).
>
> > Are there other examples besides the sphere for which the oracle calls can be computed efficiently? E.g., if the authors can talk about an example with Stiefel manifolds, that would be nice to know about.
>
> Thanks for mentioning this interesting example! Yes, the oracle calls, for instance, can also be computed over tori ($\mathbb{R}^d / \mathbb{Z}^d = [0, 1]^d$), which we did in the newly added experimental section. For the Stiefel manifold, we note that its eigenfunctions form a linear subspace of (harmonic) polynomials. Therefore, the discussion provided for spheres also applies to the Stiefel manifold, as its eigenfunctions can similarly be expressed as low-degree polynomials.

---

> > ### Author Response · Authors · 2024-12-01
> > **Additional concerns?**
> >
> > Thank you very much for your constructive review and positive feedback. We are wondering if there are any additional concerns we should address that might lead to a more positive evaluation of our work in your view.

---

### Official Review · Reviewer_SkRd · 2024-11-07

**Soundness:** 3
**Presentation:** 3
**Contribution:** 3
**Rating:** 6
**Confidence:** 4

**Summary:**

This paper proposed a computationally efficient algorithm for  learning with exact invariances over RKHS. Specifically, on one hand, the authors demonstrated that the  proposed algorithm only has polynomial-time complexity, in sharp contrast to the previous  algorithm with invariances. On the other hand, an advantage compared to KRR is that the proposed algorithm can enforce group invariances without loss in statistical convergence rate.

**Strengths:**

1. The paper is well-written and easy to follow.

2. A novel algorithm is proposed with two main advantages: computational efficiency and the enforcement of group invariances.

3. The theoretical analysis in this paper is comprehensive and rigorous. The statistical rate of the proposed algorithm  is provided, along with detailed comparisons to the related algorithm.

**Weaknesses:**

**Writing:**

The dimension $d$ first appears before it is defined. I suggest the authors introduce the definition of $d$ earlier for clarity.

In Line 144,  the labeled samples are drawn from the product manifold $\mathcal{M}\times R$ rather than from $\mathcal{M}$.


**Numerical experiments:**

While I find the theoretical aspect interesting and well-founded, it may be beneficial to conduct simple numerical experiments to support the improvement of the proposed algorithm in computational efficiency and enforcing group invariances,  while preserving the same learning rate as the original KRR.

**Questions:**

One concern I have is  about the computational complexity comparison between the proposed algorithm and the original. You claim that your algorithm reduces the time cost from the super-exponential in $d$ to the polynomial time. However, to my knowledge, the standard KRR method can only solve the low-dimensional task, where the dimension $d$ is typically small and can be neglected compared to the number of sample points.

I also note that  the previous algorithm has computational complexity of $n^2$ (if I can ignore $d$), while your algorithm lower the complexity to $n^{(2+\alpha)/(1+\alpha)}$. I suggest that the authors carefully discuss this issue.

---

> ### Author Response · Authors · 2024-11-21
> **Response to Reviewer SkRd**
>
> We express our gratitude to the reviewer for their valuable comments and feedback. In the above, we have offered a general response to the reviewers, addressing some of their common concerns about experiments. Here, we provide additional responses to the remaining questions raised.
>
> > Writing: The dimension $d$ first appears before it is defined. I suggest the authors introduce the definition of $d$ earlier for clarity.
>
> Thank you for your comment. To address it, we have added a sentence clarifying the role of $d$ in the revised version (highlighted in red, Line 133).
>
> > In Line 144, the labeled samples are drawn from the product manifold $\mathcal{M} \times R$ rather than from $\mathcal{M}$.
>
> The data points $x_i$ are sampled from $\mathcal{M}$, while the labels $y_i$ are sampled from $\mathbb{R}$. Consequently, the tuples $(x_i, y_i)$ are drawn from the product space $\mathcal{M} \times \mathbb{R}$ (Line 144).
>
>
>
> > Numerical experiments: While I find the theoretical aspect interesting and well-founded, it may be beneficial to conduct simple numerical experiments to support the improvement of the proposed algorithm in computational efficiency and enforcing group invariances, while preserving the same learning rate as the original KRR.
>
> Please refer to our general response for the experiments.
>
> > One concern I have is about the computational complexity comparison between the proposed algorithm and the original. You claim that your algorithm reduces the time cost from the super-exponential in $d$ to the polynomial time. However, to my knowledge, the standard KRR method can only solve the low-dimensional task, where the dimension $d$ is typically small and can be neglected compared to the number of sample points.
>
>
> Thanks for your valuable comment! We answer your question from two different perspectives, computational and statistical.
>
>
> From a **computational viewpoint**, in the original problem of KRR (without invariances), there are two terms contributing to the computational complexity. The dominating term is $O(n^3)$, given that the evaluation of the Gram matrix $K(.,.)$ can be done efficiently. This $O(n^3)$ complexity term arises because, once all the $n^2$ entries of the Gram matrix $K(x_i,x_j)$ for all data points $x_i, x_j$, $i,j \in [n]$, are calculated, they only need to be plugged into the closed-form solution of the KRR estimator, which involves a matrix inversion requiring $O(n^3)$ computations.
>
> The other term is the computational complexity of calculating the Gram matrix $K(.,.)$. In practice, for most of the popular kernels, this calculation is computationally efficient. For example, for dot-product kernels (e.g., Gaussian or RBF kernels, polynomial kernels, and Matern kernels for Sobolev spaces), one only needs to evaluate the inner product between two $d$-dimensional vectors, which requires only $O(d)$ time. Therefore, the time complexity required to calculate the Gram matrix is $O(n^2 d)$. Thus, this term is usually dominated by the $O(n^3)$ term whenever $n \geq d$.
>
> Putting these pieces together, as long as the kernel can be evaluated in polynomial time in $d$, one can compute the KRR estimator in polynomial time with respect to the number of samples $n$ and the input space dimension $d$. However, in learning with **exact invariances**, one additional factor comes into play: the size of the group  $|G|$, which can be superexponential in the input dimension $d$ (for example for the case of permutation invariances). For instance, if we use group averaging, for each pair of data points $x_i, x_j$, $i,j \in [n]$, we need to compute the invariant Gram matrix $K_{\operatorname{inv}}(x_i, x_j) =\dfrac{1}{|G|} \sum_{g \in G} K\big(gx_i, x_j\big)$. This step has a complexity of order $\Omega(|G| n^2)$. Compared to the case of KRR without invariances, this term dominates as long as $|G| \geq n$, which is usually the case (the number of observations $n$ does not grow beyond superexponential dependence on the input dimension $d$). Additionally, we note that, from the perspective of complexity theory, the complexity $\Omega(|G| n^2)$ is not appealing, as it is not polynomial in the parameters of problem statement, i.e., parameters $n$ and $d$.

---

> ### Author Response · Authors · 2024-11-21
> **Response to Reviewer SkRd - Cont'd**
>
> Second, from a **statistical viewpoint**, the non-asymptotic convergence guarantees for the KRR problem depend on the number of observations $n$ and the spectral properties of the chosen kernel, e.g., the spectral decay rate. In many cases, these parameters do not have a direct dependence on the dimension of the input space $d$. In many other settings, choosing a number of observations ($n$) that is polynomially large in $d$ is sufficient for small generalization errors. For example, in the case of polynomial regression of order $k$, choosing $n$ of order $O(d^k)$ leads to small estimation errors. Consequently, the computational/statistical efficiency of KRR estimators with small errors, especially in the case of learning with exact invariances, remains an interesting and valid problem in terms of both parameters $d$ and $n$.
>
> Once again, we thank the reviewer for raising this important question, and we hope the discussion above addresses their concern.
>
> > I also note that the previous algorithm has computational complexity of $n^2$ (if I can ignore $d$), while your algorithm lower the complexity to $n^{(2 + \alpha)/(1 + \alpha)}$. I suggest that the authors carefully discuss this issue.
>
> Thank you for pointing out this question. To clarify, we note that the computational complexity that we report on Theorem 1 (main result) is conditional on the oracle calls that we use to construct the estimator. More precisely, we compute the estimator in time $\mathcal{O}\big(\log^3(|G|) n^{3/(1+\alpha)} + n^{(2+\alpha)/(1+\alpha)}\big)$ using $\mathcal{O}\big( \log(|G|) n^{2/(1+\alpha)}  + n^{(2+\alpha)/(1+\alpha)}\big)$ oracle calls.
>
> Calculating the KRR estimator (without invariances) requires $n^2$ calls to an oracle that evaluates the kernel $K(\cdot, \cdot)$ at pairs of data points. Consequently, the two algorithms rely on  different types of oracles and are not directly comparable in terms of total time complexity without considering the specific algorithm used to process the oracle calls.
>
> In our proposed framework, the oracles involve evaluations of eigenfunctions and inner products of eigenfunctions. For most cases, these oracles can be answered using efficient algorithms. For example, in hyperspheres the eigenfunctions correspond to bounded-degree (harmonic) polynomials (Line 232, highlighted in red). Therefore, oracle calls can be answered in polynomial time (more efficient algorithms, such as those based on the Fast Fourier Transform (FFT), exist for computing polynomial products but for the purposes of this paper, even a naive approach suffices to ensure that oracle calls are answered in polynomial time).

---

> > ### Comment · Reviewer_SkRd · 2024-11-24
> >
> > Thank you for your response. I would keep my score and lean to accept this paper.

---

> > > ### Author Response · Authors · 2024-12-01
> > > **Additional concerns?**
> > >
> > > Thank you very much for your reply. We are wondering if there are any additional concerns we should address that might lead to a more positive evaluation of our work in your view.

---

### Author Response · Authors · 2024-11-21
**General Response and Experiments**

We appreciate the reviewers for their constructive feedback and for recognizing the technical merits and contributions of our work. During the rebuttal phase, we have revised the manuscript and highlighted the changes in red.

Below, we provide a general response to the common question about experiments raised by the reviewers.

**Experiments:** The reviewers requested the implementation of the algorithms discussed in the paper. In the revised version, we have included an experimental section to validate the practical performance of the proposed algorithm (Spec-Avg) and to compare it with Kernel Ridge Regression (KRR). The new results are presented in the updated manuscript (Appendix C, highlighted in red).

We now summarize the experimental results and refer the reviewers to the paper for further details. In our setup, we consider regression over $d$-dimensional Tori, represented as $\mathbb{T}^d = [-1,1)^d$, with a sign-invariant target function (i.e., a function invariant with respect to the group of sign transformations $G = \lbrace \pm 1\rbrace^d$, acting on $\mathbb{T}^d$ by coordinate-wise sign inversion).

We report the performance of our proposed algorithm and {KRR} for $n$ i.i.d. samples ($10 \leq n \leq 1000$) under different regularization parameters $D$ (for our algorithm) and $\lambda$ (for KRR). Our main finding is that the proposed algorithm learns the target function with a desirable test loss, comparable to the KRR estimator. Moreover, we also report an invariance discrepancy (defined in the paper, Appendix C), which measures the closeness to invariance for any estimator. For our algorithm, this value is zero, as the estimator is invariant by construction. We observe that KRR, in contrast, does not necessarily produce invariant estimations in practice.

---

### Meta-Review · Area_Chair_GzFE · 2024-12-19

**Metareview:**

This paper aims to learn with exact invariances using kernel regression over manifold input spaces. Under the assumption that the geometric properties of the
input space can be accessed, the paper proposed a new algorithm that learns a classifier with exact invariances in polynomial time, while retaining the excess population risk as the original kernel regression.

The paper is well written, and the contributions are clear.  Although the generation error is not improved by the invariance, the minimax analysis provided in the rebuttal alleviates the concern.  My major concern, however, is the applicability of the proposed method in general machine learning.  I totally understand that the paper’s major contributions are theoretical.  However, the two assumptions on the oracle appear rather demanding in real applications - what can be guaranteed when the Laplace–Beltrami operator is approximated by the graph Laplacian? The examples given in the paper are rather simplistic (unit sphere), making it quite difficult to see how the method can have a significant impact on the real practice of learning with exact invariance.

**Additional Comments On Reviewer Discussion:**

The rebuttal has been noted by the reviewers and have been taken into account by the AC in the recommendation of acceptance/rejection.

---

> ### Public Comment · ~Ashkan_Soleymani1 · 2025-02-14
> **We are surprised!!**
>
> We are surprised by this decision!
>
> Our work was placed in the **learning theory** area, so we expected it to be evaluated accordingly. As the title suggests, we focus on the **computational complexity** of learning with **exact invariances**. By "exact," we mean the underlying manifold is known, so no approximation is needed. Therefore, the Area Chair’s comment on “approximating the Laplace–Beltrami operator via the graph Laplacian” is less relevant—exact invariances are impossible if the manifold is only approximated.
>
> Our “oracle” assumption is general and naturally applies to classic manifolds such as spheres, tori, and Stiefel manifolds. We also provided concrete experiments showing that practicality of our algorithm. Additionally, **harmonic analysis**, a well-established field of mathematics, primarily studies functions on spheres. This makes the claim that our examples (e.g., the unit sphere) are “simplistic” seem unwarranted.
>
> We hope this clarifies any misunderstandings.

---

### Decision · Program_Chairs · 2025-01-22

Reject